# What Makes CLIP More Robust to Long-Tailed Pre-Training Data? A Controlled Study for Transferable Insights

**Xin Wen**[1]   **Bingchen Zhao**[2]   **Yilun Chen**[3]   **Jiangmiao Pang**[3][†]   **Xiaojuan Qi**[1][†]

[1]The University of Hong Kong   [2]University of Edinburgh   [3]Shanghai AI Laboratory

{wenxin, xjqi}@eee.hku.hk   pangjiangmiao@pjlab.org.cn

## Abstract

Severe data imbalance naturally exists among web-scale vision-language datasets. Despite this, we find CLIP pre-trained thereupon exhibits notable robustness to the data imbalance compared to supervised learning, and demonstrates significant effectiveness in learning generalizable representations. With an aim to investigate the reasons behind this finding, we conduct controlled experiments to study various underlying factors, and reveal that CLIP's pretext task forms a dynamic classification problem wherein only a subset of classes is present in training. This isolates the bias from dominant classes and implicitly balances the learning signal. Furthermore, the robustness and discriminability of CLIP improve with more descriptive language supervision, larger data scale, and broader open-world concepts, which are inaccessible to supervised learning. Our study not only uncovers the mechanisms behind CLIP's generalizability beyond data imbalance but also provides transferable insights for the research community. The findings are validated in both supervised and self-supervised learning, enabling models trained on imbalanced data to achieve CLIP-level performance on diverse recognition tasks. Code and data are available at: https://github.com/CVMI-Lab/clip-beyond-tail.

## 1 Introduction

The development of contrastive language-image pre-training (CLIP) [36, 44, 57, 68, 93] has demonstrated unprecedented success in learning generalizable representations, empowering zero-shot vision tasks and robustness to natural distributional shifts. This success can be primarily attributed to the effective use of large-scale uncurated image captioning datasets collected from the web. A recent trend involves delving into the distribution of these datasets and explicitly introducing interventions to the curation process to create better data for training [29, 91]. However, limited research has been conducted on analyzing the distribution of concepts/classes in these datasets and the behavior of CLIP under varying distributions. This work thus starts by presenting a *concept-centric* analysis of existing web-scale image-text datasets and models pre-trained accordingly (Fig. 1).

**Motivation.** Our motivation for this study arises from an intriguing observation of CLIP's zero-shot performance on ImageNet: CLIP is notably more robust to pre-trained data imbalance than supervised learning. We examine various vision-language datasets at different scales, and analyze their distribution with respect to ImageNet classes. We find that image-text datasets share an extremely imbalanced class distribution (Fig. 1a). Interestingly, we find that the zero-shot classification performance of trained CLIP models is more robust to this imbalance, especially compared to models obtained by supervised learning. This is evidenced by a weaker correlation between a class's performance and its

---

[†]Corresponding author.

38th Conference on Neural Information Processing Systems (NeurIPS 2024).

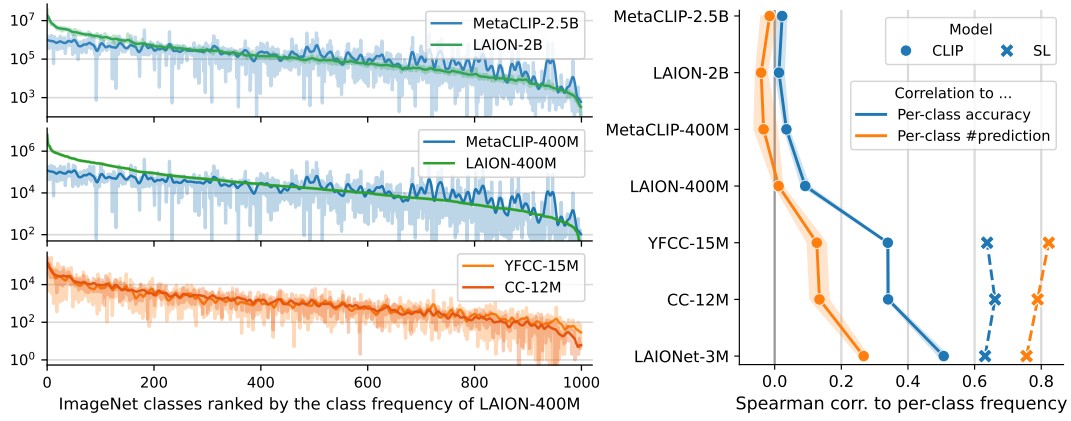

(a) Class frequencies (log scale) ranked by LAION-400M.   (b) Correlation between class-wise statistics.

Figure 1: Per-class statistics of image-text datasets and models trained on top. (a) A highly imbalanced class distribution is *shared* across datasets.[‡] (b) Compared to supervised learning (✖ SL), CLIP's performance (measured by ● accuracy) is *less biased* by data frequency, and the classifier is notably uncorrelated (measured by model's number of ● prediction per class). Besides, the correlation narrows as data scales up. Both aspects indicate implicit re-balancing mechanisms exist in CLIP.

frequency (Fig. 1b). This trend is consistent across CLIP models and pre-training datasets and even holds true for smaller-scale datasets like CC-12M [12]. This phenomenon inspires us to study the underlying causes for CLIP's relative robustness toward data imbalance and what we can learn from.

**Our study and findings.** To answer the question above, we conduct controlled experiments to analyze factors including supervision signal and pretext task (Fig. 3), data distribution (Fig. 4), scale (Fig. 5), and open-world concepts (Fig. 6). Our extensive studies have led us to the following findings:

- Language supervision, particularly the texts with increased descriptiveness (informativeness), enhances both the robustness and discriminability of CLIP, and preserves more feature variation.

- CLIP's pretext task forms dynamic classification problems, wherein only a subset of classes is present during training, effectively isolates biases to dominant classes, and balances learning signal.

- Severe data imbalance in web datasets increases the risk of bias in models. However, distribution shift and higher data diversity in them can enhance robustness, albeit a trade-off in data efficiency.

- CLIP's robustness and discriminability improve together with data scaling, benefitting from its ability to utilize open-world data, a privilege not accessible to supervised learning.

**Applications.** Inspired by the findings of our study, we found that this robustness to data imbalance can be transferred to supervised and self-supervised learning models with simple techniques by making the classification task dynamic during training. Under extremely imbalanced data scenarios, we show that a vanilla classification model can also generalize well to tail (or even open-world) classes as well as CLIP via 1) fixing the classifier with class prototypes from pre-trained CLIP text encoder, and 2) training with randomly subsampled vocabulary (results in Fig. 8, and analysis in Fig. 9). Beyond classification, we also show improved transferability on DINO [11] pre-trained on uncurated web data by simply randomly subsampling the prototypes in training (Fig. 10).

**Summary.** Our study is one of the pioneering efforts to explore CLIP's robustness in the context of imbalanced data distributions. Our exploration provides a comprehensive analysis that uncovers the mechanisms contributing to CLIP's robustness against data imbalance. As we will demonstrate in this paper, the insights gained from our research are transferable to other domains, including supervised and self-supervised learning frameworks.

---

[1]MetaCLIP [91] is relatively more balanced than other datasets due to concept re-balancing in curation.

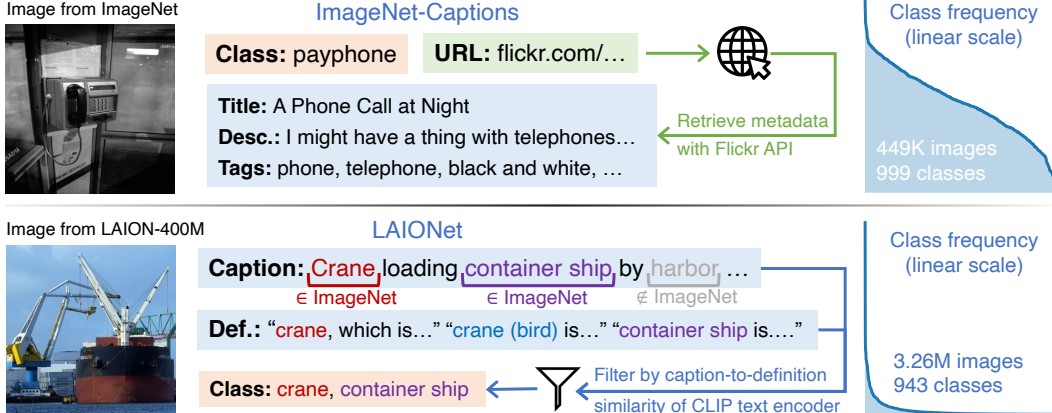

Figure 2: Curation process and distribution of datasets used in our controlled study. Top: IN-Caps [27] augments train images of ImageNet with texts by querying Flickr with image URLs. The texts include title, description, and tags. Bottom: LAIONet [77] is a filtered subset of LAION-400M [73], obtained by matching ImageNet classes with captions and filtering by CLIP text encoder for disambiguation.

## 2 Related work

**CLIP's distributional robustness.** The debut of CLIP not only set the state-of-the-art performance on conventional image classification benchmarks but also demonstrated unprecedented robustness to challenging distribution shifts. Studies have shown that this robustness stems from the diverse training distributions CLIP has seen during training time [27, 69]. Also, it is shown that the data quality plays an important role in enhancing the distributional robustness of CLIP [58]. It may seem that CLIP obtains the improvement distributional robustness due to the similarity of pretraining data to the distribution shifted data, but [55] shows that it is not the case where even after pruning similar data, CLIP still obtains strong robustness, indicating generalizable representations are learned.

**Learning from uncurated data.** Apart from robustness to distribution shifts, previous works have also delved into the nature of uncurated large-scale datasets [35, 49, 77, 91]. Studies have shown that self-supervised learning can produce more robust models than supervised learning on uncurated data [35, 49]. Moreover, focusing on learning of subsets of the entire dataset [9, 82] has shown to further enhance self-supervised learning from uncurated data. On the learning on uncurated data, the language information has shown to help learn good representations [71]. Balancing the concept distribution of uncurated data has shown to be a scalable way of learning good models [91]. However, the uncurated data is not all harmful for performance, the lower intra-class similarity of the data is shown to help preserve information/variation in representations [77], but at low data efficiency [85].

**Generalization of vision models.** One of the main themes of computer vision research in the era of deep learning is the search for more generalizable models. Works have focused on self-supervised pretraining with only images, among which contrastive learning [13] and self-distillation [11, 61] are shown to be effective. With the introduction of large-scale image-text datasets [73, 74], there is a huge interest in learning more generalizable vision representations from additional language supervision. While techniques for incorporating language supervision have been proposed [19, 36, 68, 72, 94], further exploration of how semantic grounding help improves the generalization is needed [21]. To fully utilize language supervision, using synthetic data from large language models to improve language supervision is a newly emerged research area [25, 26].

## 3 What makes CLIP more robust to long-tailed pre-training data?

In the following, we conduct a series of controlled experiments to systematically evaluate the role of various factors on the robustness of CLIP to data imbalance. These factors include supervision signal (Sec. 3.2), pretext task (Sec. 3.3), data distribution (Sec. 3.4), data scale (Sec. 3.5), and open-world concepts (Sec. 3.6). Moreover, we also provide some insights on CLIP's feature space in Sec. 3.7.

## 3.1 Setting

**Datasets.** Experiments in this study are conducted on variants of two image-text datasets: ImageNet-Captions [27] and LAIONet [77] to allow better data-centric control. An overview is shown in Fig. 2. Both datasets provide images with their paired captions, and class labels on ImageNet. The captions of ImageNet-Captions are in the format of title, description, and tags (some can be missing for a specific image), which allows control of captions' descriptiveness. Images of LAIONet are drawn from LAION, which has a higher intra-class variability and is extremely imbalanced across classes. This makes it more challenging to train on and allows isolating the effect of data distribution.

**Models.** We consider both CLIP and supervised learning (SL) with ResNet-50 as the backbone. Given that CNNs are generally considered less robust than ViTs [4], this choice also enables us to infer the robustness of other models. For SL, we align most details with CLIP [68] to rule out the effect of irrelevant factors. *E.g.*, we use the same weak data augmentation as CLIP, adopt a prototypical classification head (*i.e.*, $\ell_2$-normalizing both features and classifier weights), and apply a learnable temperature to logits. The training schedules of CLIP and SL follow [15] and [27], respectively. Models are fully trained from scratch by default. More details are provided in Appx. C.

**Metrics.** We compute Spearman correlation coefficients [78] between class frequency and models' statistics (class-wise top-1 accuracy and number of samples predicted as each class). Besides, we also consider metrics from neural collapse literature [32, 63] for analyzing feature distribution. Formally, defining the global feature mean $\boldsymbol{\mu}_G = \mathrm{Avg}_{i,c}\,\boldsymbol{h}_{i,c}$, class-level means $\boldsymbol{\mu}_c = \mathrm{Avg}_i\,\boldsymbol{h}_{i,c}$, within-class covariance $\boldsymbol{\Sigma}_W = \mathrm{Avg}_{i,c}(\boldsymbol{h}_{i,c} - \boldsymbol{\mu}_c)(\boldsymbol{h}_{i,c} - \boldsymbol{\mu}_c)^\top$, and between-class covariance $\boldsymbol{\Sigma}_B = \mathrm{Avg}_c(\boldsymbol{\mu}_c - \boldsymbol{\mu}_G)(\boldsymbol{\mu}_c - \boldsymbol{\mu}_G)^\top$, the metrics are defined as:

$$\mathrm{NC1} = \mathrm{Tr}\Big(\boldsymbol{\Sigma}_W\boldsymbol{\Sigma}_B^\dagger/C\Big), \quad \mathrm{NC2} = \mathrm{Avg}_{c,c'}\left|\frac{\boldsymbol{\mu}_c^\top\boldsymbol{\mu}_{c'}}{\|\boldsymbol{\mu}_c\|\|\boldsymbol{\mu}_{c'}\|} + \frac{1}{C-1}\right|, \tag{1}$$

where $\dagger$ denotes the Moore-Penrose pseudoinverse, $\boldsymbol{h}_{i,c}$ is the feature of the $i$-th example in class $c$, and $C$ is the total number of classes. Intuitively, NC1 and NC2 measure the compactness and separation of clusters, respectively. NC1 approaches zero when the within-class variation of features becomes negligible, and NC2 converges to zero when classifiers reach maximal and equal margins (*i.e.*, ETF structure) [63]. Note that these two metrics are originally defined as an average across classes, and it is simple to obtain per-class NC1 and NC2 metrics, measuring the variability of *a specific class* or its average margin to all other classes.

## 3.2 (Descriptive) language as supervision signal

**Setting.** We start by examining the impact of language supervision, the primary distinction between CLIP and other contrastive learning approaches. This is done by creating *texts with roughly monotonic*

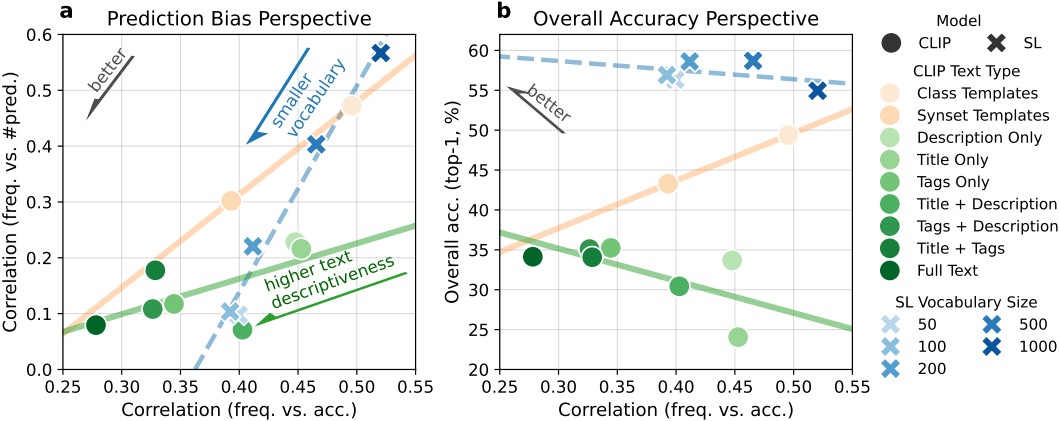

Figure 3: Results on IN-Caps about ● text descriptiveness and ✖ vocabulary size. 1) Increasing ● text descriptiveness improves both robustness (a) and discriminability (b) of CLIP, but the tendency varies if using ● less descriptive (template-based) supervision. 2) The gap between SL and CLIP (a) implies CLIP re-balances predictions, which is replicable by ✖ subsampling the vocabulary SL trains with.

*increasing descriptiveness* given metadata of ImageNet-Captions. For the low-diversity texts, we create ● synthetic class-centric texts using classification templates from CLIP [68] given class names or synset [56]. The ● natural language-based texts are created by concatenating different types of captions (see Fig. 2), and the descriptiveness of language supervision is controlled by the number of text types used. More details are available in Appx. C.2.

**Results.** Fig. 3 provide a comprehensive comparison between model variants from different perspectives. Restricting our view to CLIP models in the first two subfigures, ● higher text descriptiveness results in improvements in both robustness and discriminability of CLIP, as shown by lower correlation (Fig. 3a) and higher overall accuracy (Fig. 3b, $y$-axis). On the other hand, ● relatively less descriptive texts show weaker results that are close to results of ● templated-based CLIP (Fig. 3a, $x$-axis). We see this as less descriptive texts could collapse to class-centric supervision without much additional variance. Despite this, predictions of ● template-based CLIP are still notably less biased by pre-training data than ✖ SL (Fig. 3b), indicating other factors may re-balance CLIP's predictions.

### 3.3 Dynamic classification (using subsampled vocabulary) as pretext task

**Setting.** We note that the pretext of ● template-based CLIP still differs from ✖ SL. Although both formed as discrimination tasks, the vocabulary (classes in a mini-batch) of CLIP is much smaller than SL (all classes). Take using a batch size of 1024 for instance, after deduplication, the vocabulary only contains around 600 classes (for ImageNet-Captions). If negative samples are not shared across devices, the vocabulary received by each GPU can be even smaller. In contrast, the vocabulary of SL is consistent: 1000 classes for ImageNet. Considering CLIP sees far more than 1000 classes from a web-crawled dataset, the *portion* that CLIP's training vocabulary takes is even smaller. To isolate the influence of training vocabulary, we experiment by forming dynamic classifiers during SL training. This is done by randomly subsampling the vocabulary (candidate classes) to a smaller size during training, thus forming dynamic classification tasks similar to CLIP (see details in Appx. C.3).

**Results.** As shown in Fig. 3a, sampling a ✖ smaller vocabulary notably reduces SL's prediction bias, and obtains robustness similar to ● template-based CLIP. Regarding the favorable vocabulary size, smaller ones are more effective in reducing prediction bias (Fig. 3a), and intermediate ones also improve accuracy (Fig. 3b). The preferred vocabulary size for ImageNet-Captions is around 100.

**Discussion.** Our intuition of the phenomena above is that dynamic classification in some way achieves class-level re-balancing. When the ground truth (GT) corresponds to a tail class, a small vocabulary isolates the negative impact of most head classes, avoiding bias towards them and enabling the model to focus on classifying the tail class itself. Besides, it is worth noting that as demonstrated in [32, 50], optimization continues after the model's predictions reach zero error, and seeks minimum intra-class variability and maximum inter-class margin (especially larger margin around head classes). Thus when the GT is a head class, this approach limits the number of negative classes and could prevent the model from excessively distorting the representations of them through over-optimization.

### 3.4 Data distribution (level of imbalance, web distribution shift, and intra-class diversity)

**Motivation.** Motivated by the findings of [27] regarding the impact of image distribution on CLIP's robustness to natural distribution shifts, our study also examines its influence on robustness to data imbalance. As shown in [77], a higher filter threshold leads to a more condensed image distribution, a result that is confirmed in Fig. 4a. We thus create LAIONet variants of different intra-class variations by adjusting this threshold. All variants in this section keep the data scale the same as ImageNet-Captions (0.45M). In addition, due to the disparity in class distribution between LAIONet and ImageNet-Captions, we also create a variant that aligns with the class frequencies of ImageNet-Captions ('=freq') while preserving web image distribution. This variant is sampled from the full version (3.26M) that uses a threshold of 0.7. More details about datasets are provided in Appx. C.5.

**Results.** A comparison between models trained on the aforementioned datasets is present in Fig. 4b. We find that web data is not naturally friendly for de-biasing, but could have made models more biased due to extreme data imbalance (comparing '=freq' with other columns). The distribution shift of web data could improve robustness if a ● pre-trained text head is available (circles *vs.* squares, last column). If not, scaling may help. Moreover, results with smaller thresholds also turn out to be more robust, indicating that higher intra-class data diversity (smaller threshold) improves robustness.

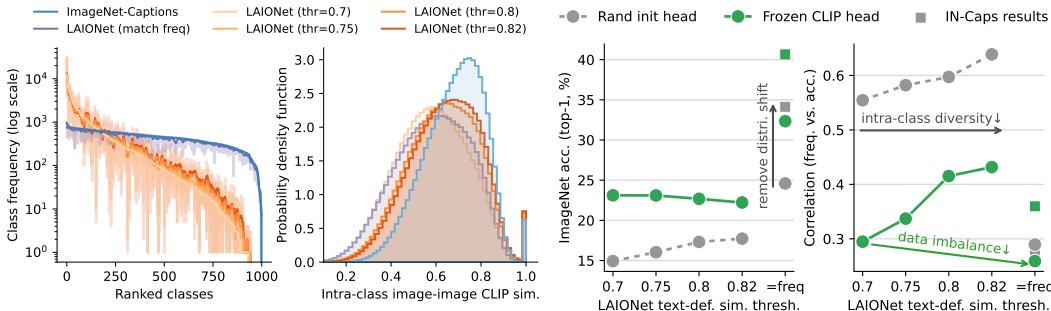

(a) Distrib. of LAIONet variants (same scale as IN-Caps). (b) Results of CLIP trained on LAIONet variants.

Figure 4: Results on LAIONet about data distribution (level of data imbalance, distribution shift, and data diversity). 1) Extreme data imbalance makes models more prone to bias (last column *vs.* others). 2) Distribution shift (●● *vs.* ■■, last column) harms discriminability but could improve robustness if pre-trained text head is used. 3) Higher data diversity (smaller threshold) also improves robustness.

## 3.5 Data scaling (also achievable via language pre-training)

**Motivation.** We note that the correlations of CLIP in Fig. 3a ($x$-axis) are still higher than that of open-source models in Fig. 1b. One key remaining factor is the scale of pre-training data (see Fig. 1b for large-scale results). Given that ImageNet-Captions is small-scaled (see Fig. 2), experiments following are conducted on LAIONet. See Appxs. C.4 and C.5 for more details about the setting.

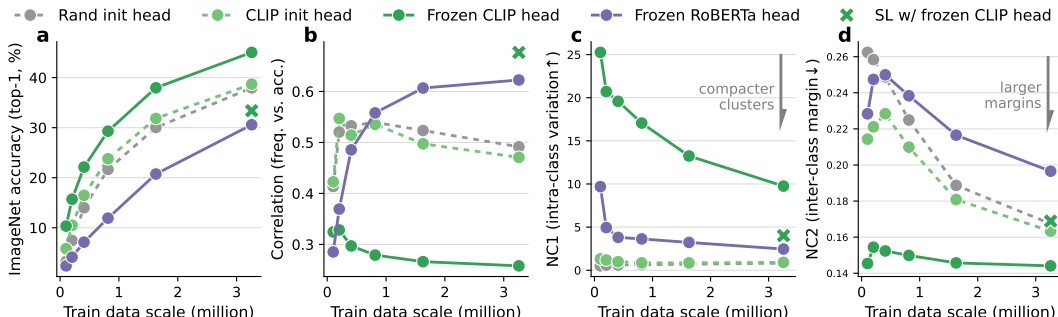

Figure 5: Results on LAIONet subsets about data scale and text encoder. 1) CLIP's discriminability (a) and robustness (b) co-improve as data scales up, and can be boosted by pre-trained heads. 2) A frozen head helps CLIP preserve intra-class variation (c) while not harming margins (d), which can be lost if fine-tuned. It is also unattainable by SL even using the same head. 3) Language pre-training using CLIP is more favorable for image-text tasks than pure language modeling (*e.g.*, RoBERTa [51]).

**Results.** Fig. 5 presents the results obtained from uniformly subsampled subsets of LAIONet. These findings extend the scaling law: as data scales, ImageNet zero-shot accuracy (Fig. 5a) and models' robustness to data imbalance (Fig. 5b) improve simultaneously. We also provide a comparison between text encoders: ● training from scratch, initializing with ● pre-trained CLIP (frozen) or ● frozen RoBERTa [51], or ● fine-tuning the text encoder together. ● Frozen CLIP language head enables the vision model to leverage a well-established feature space as supervision, achieving better data efficiency (Fig. 5a) and robustness to data imbalance (Fig. 5b). ● Fine-tuning CLIP text head results in over-fitting (similar results with ● training from scratch), and ● RoBERTa does not suit the contrastive task and adversarially affects performance. Further investigation through NC-based metrics shows ●● frozen heads effectively preserves intra-class variation (Fig. 5c), which is at risk of being lost when ● fine-tuned. Both ● frozen and ● fine-tuned heads contribute to inter-class margins (Fig. 5d), and if ● randomly initialized, scaling training data still can achieve improved margins. Compared to ✖ SL, CLIP can better utilize web-crawled data and pre-trained text encoder (Fig. 5a). But note that when evaluating close-set accuracy, the data efficiency of CLIP is still much lower than SL trained on classification datasets (*e.g.*, ImageNet).

## 3.6 Utilization of open-world concepts

**Motivation.** One overlooked factor in Sec. 3.5 (on 1K ImageNet classes) is the existence of massive open-world concepts in web-crawled datasets. CLIP only requires weak image-text supervision and is thus not bound by a pre-defined vocabulary. The open-world concepts may share useful information with close-set ones and generalization could happen when data scales up. This section presents experiments on ImageNet-Captions and YFCC-15M subsets that reveal scaling effects of the number of concepts/classes. Results are shown in Fig. 6 and details of datasets can be found in Appx. C.5.

**Results.** We present results on ImageNet-Captions subsets (evaluate on 100 classes) and YFCC-15M subsets (evaluate on 1K classes) in Fig. 6 to validate this. IN-Caps-100 stands for a 100-class subset of ImageNet-Captions, and IN-Caps (10%) denote a 1K-class subset at the same scale as IN-Caps-100. In Fig. 6a, both SL and CLIP attain additional robustness from the scaling of concept and data. However, expanding the vocabulary for SL is label-expensive in practice. Thus concepts other than ImageNet classes in YFCC-15M do not benefit SL in Fig. 6b.

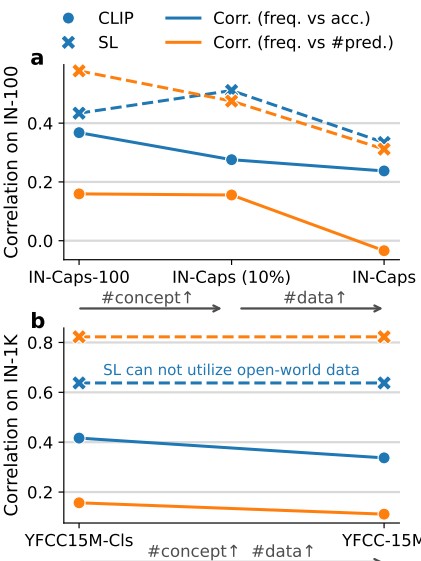

Figure 6: CLIP can benefit from open-world concepts. (a) Train on IN-Caps variants, and evaluate on 100 classes. (b) Train on YFCC-15M variants, and evaluate on 1K classes.

## 3.7 Understanding the feature distribution of CLIP pre-trained at scale

**Setting.** The results above have shown that the discriminability and robustness to data imbalance improve simultaneously as pre-training data scales up (Sec. 3.5). Then if pre-trained on sufficient data, when does CLIP fail (Fig. 7a.1), what does data imbalance affect (Fig. 7a.2), and how are they reflected in the feature space (Fig. 7b)? To answer these questions, we consider 3 vision feature-related metrics ($\bullet$ NC1, $\bullet$ NC2$_M$, $\bullet$ NC2$_M^{nn}$) and 2 text feature-related metrics ($\bullet$ NC2$_W$, $\bullet$ NC2$_W^{nn}$). NC2$_M$ uses vision feature centers, and NC2$_W$ takes CLIP's text classifier as feature centers. Margins are computed as average over all other classes for NC2, and that to the nearest neighbor for NC2$^{nn}$.

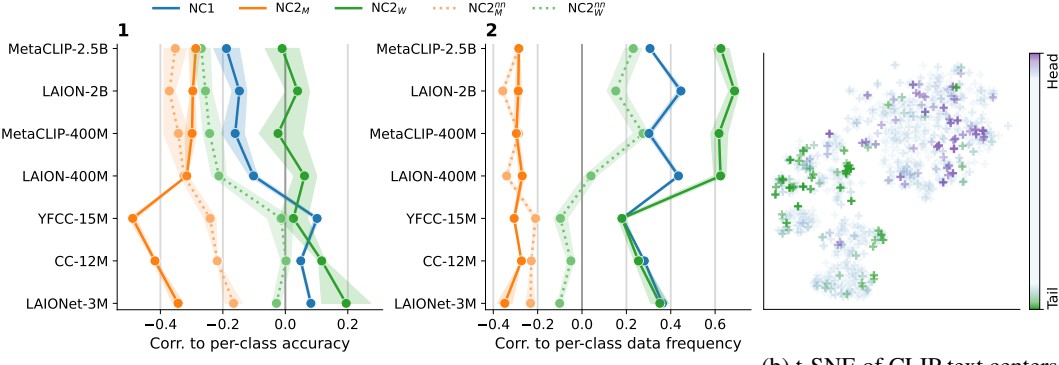

(a) Correlation between model (left), data (right), and feature statistics.

(b) t-SNE of CLIP text centers (pre-train on LAION-400M).

Figure 7: Inspecting CLIP's failures and effects of data imbalance from NC-based metrics. 1) Fail classes of smaller-scale models (12/15M) are hardly discriminative to most classes, while larger-scale models ($\geq$ 400M) only fail on some nearest-neighbor classes. 2) Data imbalance is weakly correlated with most feature statistics except NC2$_W$, denoting denser head and coarser tail classes in text space.

**Results.** Cluster compactness ($\bullet$ NC1) does not show a strong correlation with CLIP's failures (Fig. 7a.1), and the frequent classes of LAION models tend to preserve more intra-class variation (Fig. 7a.2). Besides, there are some implications from the margin between class centers ($\bullet\bullet$ NC2).

For example, Fig. 7a.1 shows that the fail classes of smaller-scale models (12/15M) are hardly discriminative to most classes (• $NC2_M$), while larger-scale models ($\geq$ 400M) only fail on some nearest-neighbor classes (• $NC2_M^{nn}$). This indicates that the failing classes already have good separation from most other classes, and the confusion primarily comes from very few hard classes. Regarding the effects of data imbalance on CLIP (Fig. 7a.2), we find a strong connection to • $NC2_W$, denoting denser head and coarser tail classes in text space. t-SNE [86] of the class centers is provided in Fig. 7b for reference, and more visualizations of vision features can be found in Fig. 20.

**Discussions.** Though weakly correlated to the class frequency, CLIP's performance is still highly biased [87, 99]. If data imbalance is not the main cause, then what are other suspect of CLIP's failures? We hypothesize that ImageNet is intrinsically biased. The classes are not of equal difficulty [17] and some are even ambiguous [6, 39, 75], *e.g.*, "sunglass" *vs.* "sunglasses". In this case, it is possible for a model trained on the balanced ImageNet to be biased [17], and some errors are unsolvable no matter how much training data is added. Besides, CLIP leverages open-world concepts in training, which are not counted for frequency but still could affect close-set performance. Moreover, such biases might be connected with CLIP's hallucination [31, 53, 92]. We believe these are valuable questions to be explored. In supplement to this discussion, we also discuss CLIP's bias measured on broader sets of concepts in Appx. A.2 and the effects of data imbalance on CLIP in Appx. A.5.

## 4 Acquiring CLIP-level generalization

This section shows findings from CLIP's underlying mechanisms can be applied to both supervised learning (Sec. 4.1) and self-supervised learning (Sec. 4.2) under severe data imbalance.

### 4.1 Data-imbalanced learning: an extreme case

In quest of the limit of CLIP's robustness to pre-training data imbalance, we create an extreme case based on ImageNet-Captions: trimming the tail classes to only one shot, or even completely zero shot (*i.e.*, an open-world setting). We then train models on this trimmed dataset, and evaluate performance on ImageNet regarding tail/other classes. As shown in Fig. 8, at the scale of ImageNet-Captions ($\sim$0.45M), • CLIP trained from scratch also fails on tail classes when trained under severe data imbalance. Despite this, by adopting a • pre-trained text encoder following Sec. 3.5, CLIP acquires open-world knowledge and demonstrates superior generalization on tail (and open-world) classes. Then how much can an SL model acquire such generalization? Surprisingly, we find training it with ✖ frozen class prototypes produced by CLIP text head is not effective. Instead, also ✖ subsampling the vocabulary during training is necessary to achieve a similar level of generalization as CLIP.

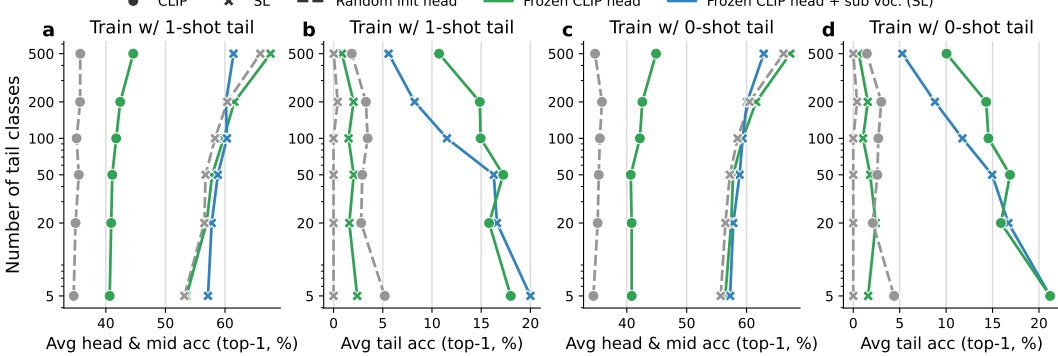

Figure 8: An extreme case: we train SL models on IN-Caps variants that have tail classes trimmed to only one shot (a & b) or even zero shot (c & d), and evaluate the accuracy on the tail and other classes. • CLIP with a frozen pre-trained text encoder shows superior generalization, which can be acquired by a ✖ SL model with ✖ fixed class prototypes from CLIP and ✖ vocabulary subsampling.

To understand the underlying mechanisms, we present a case study on the affinity matrix between classifiers, and tail class accuracies under the zero-shot tail (50 classes) setting in Fig. 9. The affinity matrices of the classification head (see Fig. 9a, we subsample 100 classes for visualization) demonstrate that the learned tail prototypes collapse to singularity, while the class prototypes from

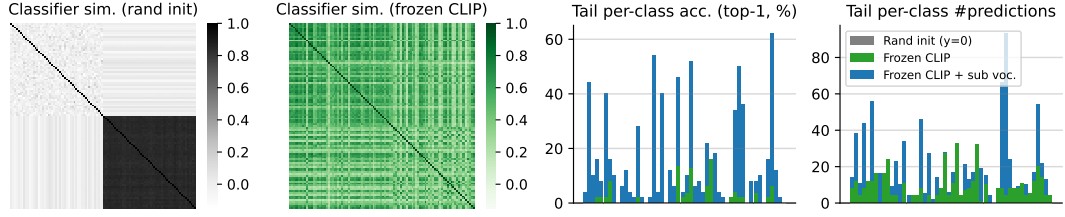

(a) Affinity matrices of the classification head.

(b) Distributions of models' per-class statistics.

Figure 9: A case study of SL under the zero-shot tail setting. (a) SL models seek maximal margins between classifiers, and tail prototypes collapse together. Instead, CLIP has a healthier structure. (b) Using CLIP head solely is less effective, and voc. subsampling is needed for CLIP-like generalization.

CLIP maintain a healthier structure. Replacing the learned head with frozen CLIP prototypes alleviates classifier bias. However, per-class accuracies (see Fig. 9b) show that using this head alone is merely effective, only small improvements are observed in very few classes, indicating that the representations are still biased. Additionally, applying vocabulary subsampling overcomes the hidden bias in supervision, allows the representations to fit the manifold encoded by CLIP text embeddings, and generalizes to open classes that CLIP has seen in pre-training. We note that this setting shares similarities with open-vocabulary recognition. Surprisingly, we indeed find a similar technique (termed federated loss) used in open-vocabulary object detection (OVOD) [98], but few explorations exist in the relevant literature. Our study provides a thorough analysis of this technique from another perspective, and we hope it can motivate future applications in this field.

### 4.2 Empowering self-supervised learning in-the-wild at scale

To show the universality of the aforementioned techniques, we also explore the application in improving self-supervised learning when pre-trained on imbalanced data. As discussed in [3, 61], DINO's performance is sensitive to the imbalance in web-crawled pre-training data, and thus data deduplication is a crucial process in DINOv2 [61]. As discussed by a recent study [30], the learnable prototypes of DINO (akin to the classifier of SL) may be biased to imbalanced data and many collapses (like Fig. 9a). We hypothesize that applying subsampling to the prototypes may alleviate this phenomenon. Our intuition is that the operation resembles dropout and could encourage better utilization of the online-learned prototypes of DINO, thus improving representations learned from uncurated web data. Based on vanilla DINO [11], we randomly subsample prototypes (instead of using them all) during the calculation of the self-distillation loss (see details in Appx. D). All models are pre-trained for 100 epochs on LAIONet, and evaluated on the transfer learning benchmark of [40].

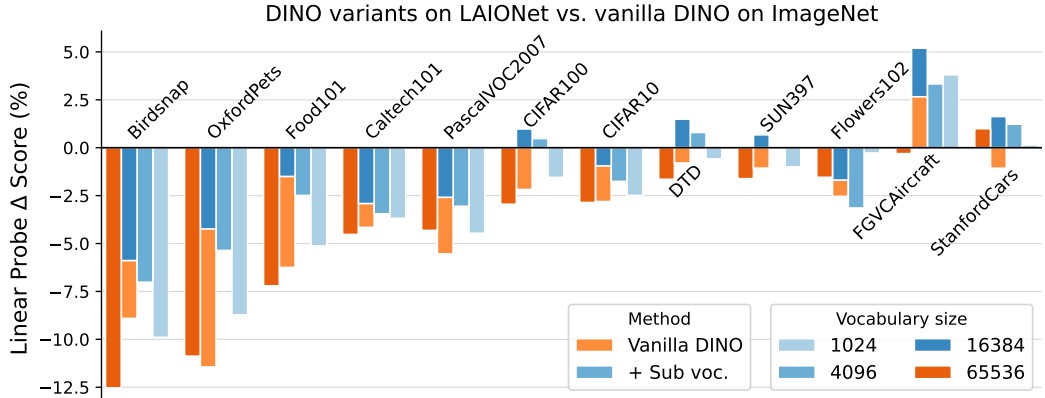

Figure 10: Transfer learning results of DINO variants pre-trained on LAIONet *vs.* vanilla DINO trained on ImageNet. Extreme data imbalance makes LAIONet much harder for DINO to learn transferrable representations. The ■ vocabulary subsampling strategy effectively helps ■ DINO alleviate such defects and generally match ImageNet-pretrained performance.

Results in Fig. 10 and Tab. 2 show that compared to pre-training on ImageNet, ■ vanilla DINO's performance drops notably among 11 datasets out of 12. Instead, ■ vocabulary-subsampling narrows the gap by a large margin, highlighting this technique's effectiveness on large-scale data in the wild. To rule out the influence of total vocabulary size (number of prototypes), we also train ■ vanilla DINO with reduced vocabulary (16384). This model is notably weaker than that trained with ■ subsampling (16384 for each training iter, 65536 in total), and supports the improvement's effectiveness.

## 5 Limitations, future work, and broader impacts

**Limitations.** Our study has focused on the robustness of CLIP-type models in relation to the data imbalance naturally raised from web data sources. We have demonstrated that our findings are transferrable to the supervised and self-supervised learning setting for classification tasks. However, we acknowledge that our estimation of image-text datasets' concept frequency is based on a simple rule-based pipeline, which could be prone to caption noise, multi-label, and ambiguity. Besides, CLIP models are not only employed for classification tasks, the study of leveraging CLIP for open-world detection or segmentation is the area our study does not cover. Additionally, given the nature of the web-based data sources used in our study, we acknowledge that the data may contain implicit bias or harmful information. We provide more discussions in Appx. A.

**Future work.** Our findings cover insights in language supervision, pretext task, data scaling, and concept scaling, but only a small portion are validated in application. One direction for future work is to explore the use of language supervision and open-world data in recognition models. Besides, a recent work [43] finds Adam optimizer to outperform (stochastic) gradient descent on heavy-tailed data, which can be another factor in CLIP's robustness and is worth further exploration. On the other hand, we are interested in extending our discovery to the open-world detection and segmentation tasks to see if our findings still hold under these more challenging scenarios.

Furthermore, as we have analyzed in our study, language supervision plays an important role in achieving such robustness to the data imbalance, thus we are also interested in studying whether or not similar traces of generalization exist in (multi-modal) large language models (*e.g.*, Llama [83], BLIP-2 [45], LLaVA [48], *etc*.). However, despite being trained on large-scale data with language supervision, recent works show that LLM/MLLMs still suffer from long-tailed training data [37, 46], and their performance is highly correlated with the frequency that corresponding knowledge appeared in training [1, 95]. This indicates that generative models might be intrinsically more prone to long-tailed data than contrastive models like CLIP, and injecting rebalancing mechanisms into the generative process could be valuable for future explorations.

**Broader impacts.** We provide an in-depth analysis of CLIP's robustness to data imbalance, which helps understand the effectiveness of CLIP. The techniques here are also shown to be effective for other domains (supervised learning and self-supervised learning) to overcome biases in tail under-represented classes. Thus, we expect our work not to pose potential negative societal consequences but rather to improve society's overall equality and inclusiveness.

## 6 Concluding remarks

Our work starts with the observation that although web-crawled datasets share an extremely imbalanced data distribution, CLIP is relatively more robust to it. Extensive studies on 1) language supervision, 2) pretext task, 3) web data distribution, 4) data scaling, and 5) open-world concepts reveal significant findings about the underlying mechanisms of this robustness. We have also demonstrated that these findings can be transferred to classification and self-supervised learning methods, yielding improved generalization under pre-training data imbalance. Our study uncovers key factors of CLIP's robustness to pre-training data imbalance, and provides new perspectives to understand its generalizability. The insights learned are validated on tasks from extremely long-tailed supervised learning to self-supervised learning on web-crawled data. While CLIP has been a game changer in these research fields, it has long been utilized as is. Our study, instead, delved into the mechanisms behind CLIP, providing an opportunity to improve downstream tasks by leveraging the underlying mechanisms rather than relying solely on the model itself, with greater flexibility and adaptability.

## Acknowledgments

This work has been supported by Hong Kong Research Grant Council — Early Career Scheme (Grant No. 27209621), General Research Fund Scheme (Grant No. 17202422), and RGC Research Matching Grant Scheme (RMGS). Part of the described research work is conducted in the JC STEM Lab of Robotics for Soft Materials funded by The Hong Kong Jockey Club Charities Trust.

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

# What Makes CLIP More Robust to Long-tailed Pre-training Data? A Controlled Study for Transferable Insights (Supplementary Material)

**Xin Wen**[1]  **Bingchen Zhao**[2]  **Yilun Chen**[3]  **Jiangmiao Pang**[3†]  **Xiaojuan Qi**[1†]
[1]The University of Hong Kong  [2]University of Edinburgh  [3]Shanghai AI Laboratory
{wenxin, xjqi}@eee.hku.hk  pangjiangmiao@pjlab.org.cn

## Contents

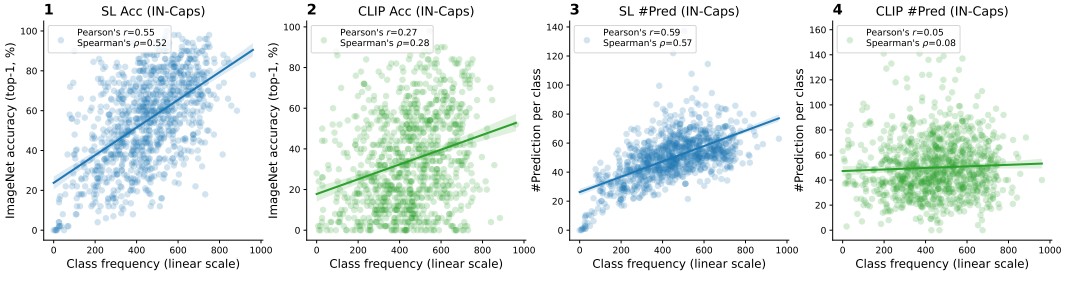

(a) Correlation statistics of models pre-trained on ImageNet-Captions.

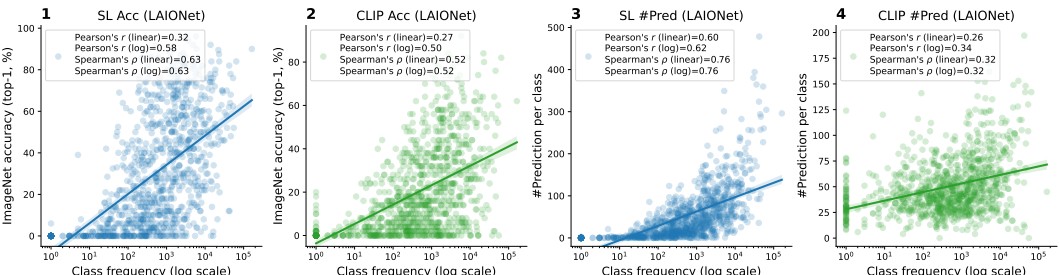

(b) Correlation statistics of models pre-trained on LAIONet.

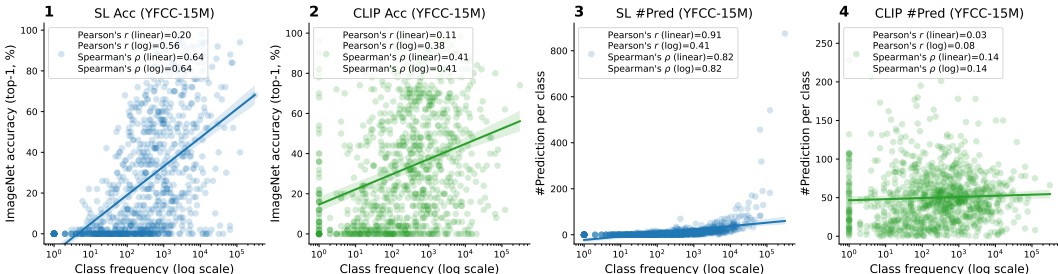

(c) Correlation statistics of models pre-trained on YFCC-15M.

Figure 11: Which is a better indicator for per-class statistics? (a) For less imbalanced IN-Caps, both Pearson's $r$ [66] and Spearman's $\rho$ [78] can model the correlation between statistics well. (b & c) For extremely imbalanced datasets (*e.g.*, LAIONet, YFCC-15M, and other web datasets), Peason's $r$ may fail even if class frequencies are processed to log scale. In contrast, Spearman's $\rho$ remains robust.

## A  Extended discussions

### A.1  What makes a good correlation indicator for per-class statistics?

Per-class statistics, especially class frequency data, can be of different levels of imbalance. A good correlation indicator should remain robust to the changes in imbalance levels and faithfully reflect the correlation between statistics. The commonly used Pearson correlation coefficient [66] ($r$) does not fit this criterion. We consider three datasets in this discussion: ImageNet-Captions, LAIONet, and YFCC-15M, which have increasing levels of data imbalance. As shown in Fig. 11, Pearson's $r$ can model moderate imbalance like ImageNet-Captions, high imbalance like LAIONet if processing the frequencies to log scale, but can fail if an extreme imbalance is met (*e.g.*, Fig. 11c.2). In contrast, the Spearman correlation coefficient [78] ($\rho$, defined as Pearson's $r$ applied to data ranks) remains robust across scenarios. We thus take Spearman's $\rho$ as the default correlation indicator used in this paper.

### A.2  Correlation statistics on broader sets of concepts

Results in the main paper only consider the distribution of concepts/classes in ImageNet-1K. In this discussion, we also consider the concept sets of broader datasets, including CUB [88], Food-101 [7],

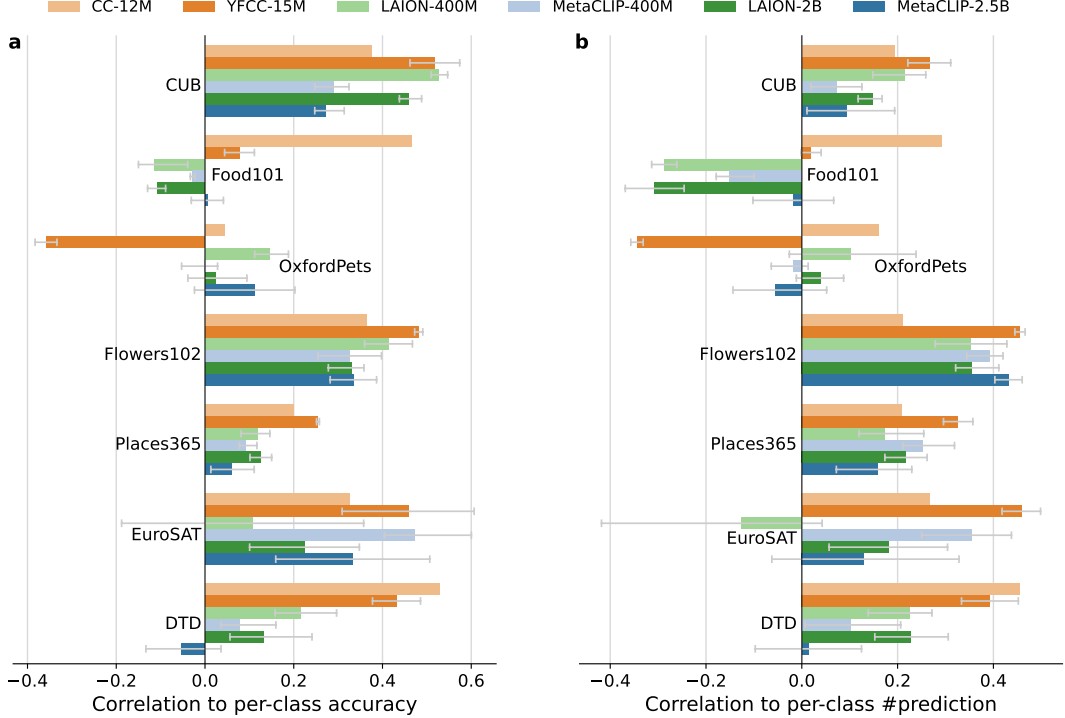

Figure 12: Correlation statistics of CLIP evaluated on broader sets of concepts. Models pre-trained at scale ($\geq$ 400M) remain robust on most datasets except fine-trained (*e.g.*, CUB and Flowers) and domain-specific ones (*e.g.*, EuroSAT). These data might be relatively rare on the web or have significant gaps with other data, thus hard to benefit from scaling or generalization from existing data.

Oxford-IIIT Pets [65], Flowers-102 [59], Places365 [96], EuroSAT [34], and Describable Textures (DTD) [16]. Pre-trained CLIP models' correlation statistics on these concept sets are as shown in Fig. 12. Models pre-trained at scale ($\geq$ 400M) remain robust on most datasets. However, some fine-trained (*e.g.*, CUB and Flowers-102) and domain-specific (*e.g.*, EuroSAT) datasets tend to be harder to learn and easier to bias. These data might be relatively rare on the web and can have significant gaps with other data formats (satellite images are relatively uncommon), thus hard to benefit from scaling or generalization from existing data.

### A.3 Distributional convergence of large-scale image-text datasets

Fig. 1a in the main paper has illustrated qualitatively that the class distributions of large-scale image-text datasets are roughly shared (correlated). Here, we also provide quantitative results about the correlation coefficients between the class distribution of different image-text datasets Fig. 13. Under most concept sets, the correlation is high and supports our claim: there exists a distributional convergence across large-scale image-text datasets. Results of MetaCLIP [91] variants are relatively less correlated, which might be due to the re-balancing operation in the curation process.

### A.4 Concept frequency estimation compared to concurrent work

Our estimation of concept frequency is based on a simple rule-based pipeline (see details in Appx. B.2), which could be prone to caption noise, multi-label, and ambiguity. A concurrent work by Parashar et al. [64] finds concept synonyms using ChatGPT [60], and estimates the class frequencies of each caption using Llama 2 [84]. These advanced techniques may produce more accurate class frequencies. In Fig. 14, we provide the correlation coefficients between our estimations and the results of [64]. The high correlation across most datasets implies an agreement and verifies the validity of our estimations. There is an exception for DTD [16], in which class names are about descriptive textures. This is more abstract than natural concepts and can be more semantically ambiguous [64], and require more sophisticated designs in frequency estimation.

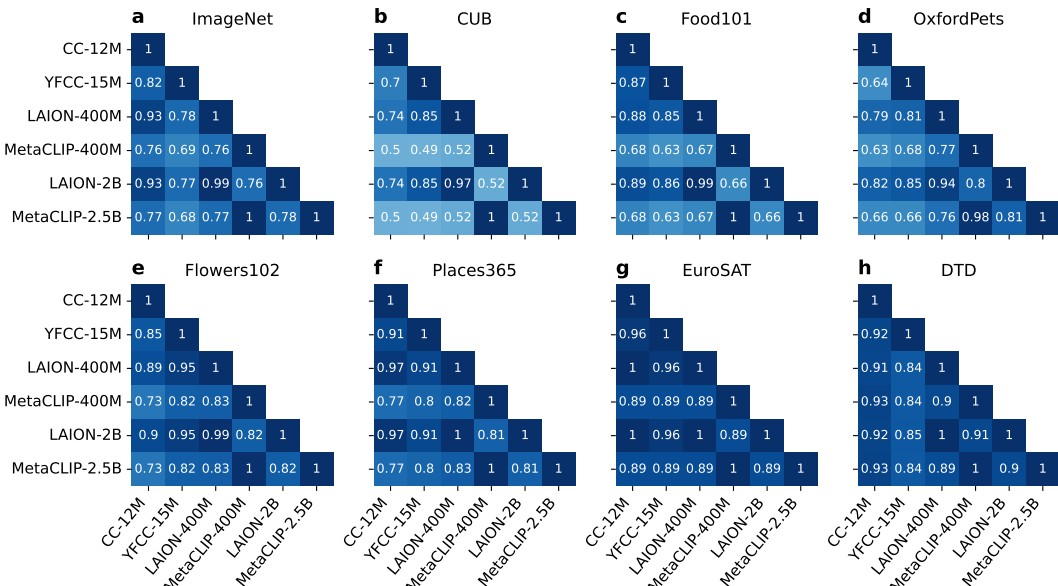

Figure 13: Correlation between class frequency statistics of different pre-training datasets under different concept sets. There is a convergence of data distribution over large-scale image-text datasets.

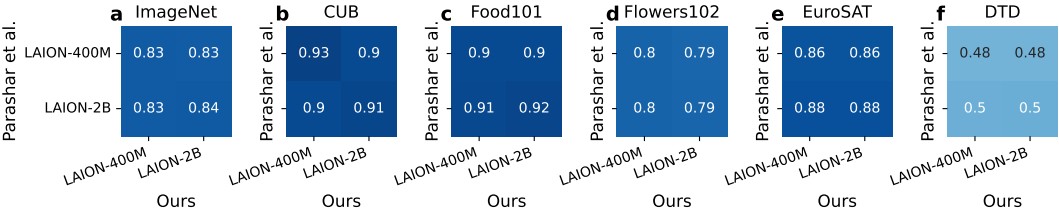

Figure 14: Correlation between class frequency statistics of our estimations and concurrent results of Parashar et al. [64]. There is an agreement on most concept sets except DTD [16], which is about descriptive textures and can be more semantically ambiguous [64].

### A.5 Is data imbalance not a concern for CLIP?

As illustrated in Figs. 1b and 5, the discriminability and robustness to pre-training data imbalance improve simultaneously as data scales up. But neither does it mean CLIP is unbiased (see discussions in Sec. 3.7), nor does it indicate CLIP is absolutely robust to data imbalance. In Fig. 15, we plot binned results of CLIP following Parashar et al. [64]. Looking at the average trend, the tail classes are still of inferior performance. However, note that the standard deviation is high, indicating there are still many good-performing tail classes. Moreover, the figure also verifies CLIP is more robust than SL (Fig. 15a), and the harm of data imbalance alleviates as data scales up (Fig. 15b).

### A.6 Motivation behind the choice of factors to study

The design of our study is largely influenced by [27], which is among the first to study data's effect on CLIP's robustness. After ruling out the effects of language supervision and data distribution, we found there is still a notable gap between CLIP and SL in Fig. 3. We then exhausted every factor we could to align details between models, and finally found the pretext task of dynamic classification to be a key factor, which could implicitly de-bias classifiers, and reproducible by applying vocabulary subsampling. Besides that, we also considered other factors like the architecture of vision backbone and text backbone, vision pre-training, stronger data augmentation, larger batch size, and test-time prompts, and did not find noticeable effects. Additionally, we looked into the properties of the dataset instead of models and found that web data had mixed effects. Further, we extend the scaling law of CLIP and find open-world data to be an effective factor.

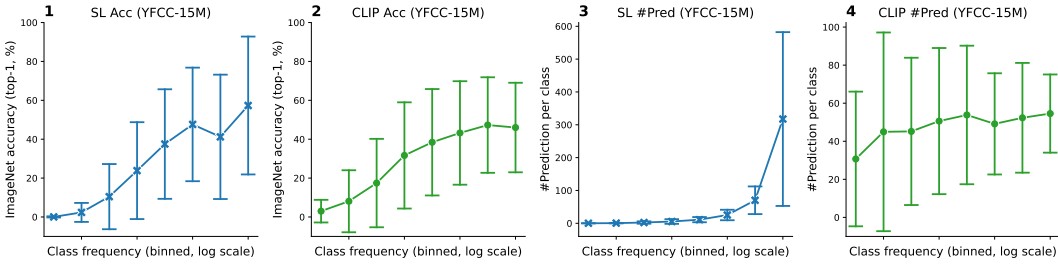

(a) Binned statistics of models pre-trained on YFCC-15M (avg ± std).

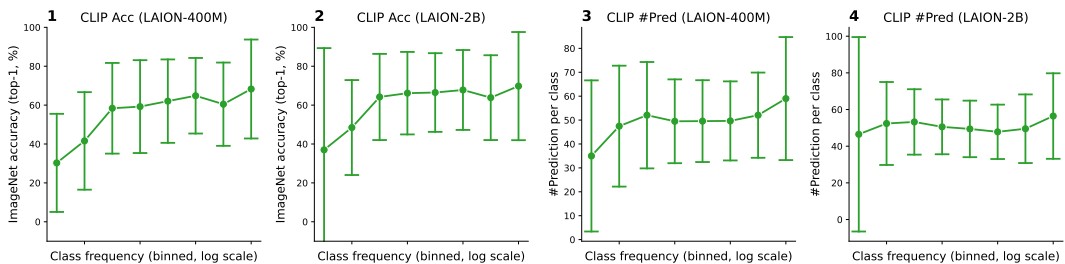

(b) Binned statistics of models pre-trained on LAION-400M and LAION-2B (avg ± std).

Figure 15: CLIP can still be biased by pre-training data. It is relatively more robust than SL (a), and the bias reduces to some extent as data scales (b *vs.* a), but the tail classes still underperform.

### A.7 Can CLIP achieve robust generalization to extremely rare concepts?

We indeed observe many. For example, among 1K ImageNet classes, 29 classes appear in YFCC-15M less than 10 times, 20 classes appear less than 5 times, 6 classes appear 1 time, and 2 classes do not appear. Within these classes, CLIP trained accordingly from scratch has $\geq 50\%$ accuracy on 12 classes. We provide detailed statistics in Tab. 1.

Table 1: Results of the tail classes on YFCC-15M.

| Freq. | 9 | 9 | 8 | 6 | 5 | 5 | 5 | 5 | 5 | 4 | 4 | 4 | 4 | 3 | 3 | 3 | 2 | 2 | 2 | 2 | 2 | 1 | 1 | 1 | 1 | 1 | 1 | 0 | 0 |
|---|---|---|---|---|---|---|---|---|---|---|---|---|---|---|---|---|---|---|---|---|---|---|---|---|---|---|---|---|---|
| Acc. | 46 | 34 | 44 | 98 | 14 | 10 | 44 | 48 | 42 | 62 | 52 | 50 | 90 | 88 | 94 | 100 | 16 | 12 | 50 | 22 | 74 | 82 | 0 | 44 | 72 | 22 | 6 | 48 | 20 |

The names of last-5 classes are: "potter's wheel", "Sussex Spaniel", "Curly-coated Retriever", "Kuvasz", and "Dandie Dinmont Terrier". We note that although our frequency calculation tries to maximize recall (*e.g.*, matches class names to captions as bag-of-words, and uses synsets), there may still be cases missed by us. Nevertheless, the results verify CLIP as a good few-shot learner.

Besides YFCC-15M, we also provide examples of LAION-400M and MetaCLIP-400M in Fig. 17.

## B Details about class frequency estimation

### B.1 Preliminaries

**Contrastive Language-Image Pre-training (CLIP).** Taking paired image-caption data as input, the pretext task is formulated as a cross-modal contrastive learning task that discriminates the paired text from a large batch of negative samples, and vice versa. After early explorations [19, 72, 94], emergent performance in representation learning, zero-shot evaluation, and distributional robustness was achieved by CLIP [68] and ALIGN [36] through training on datasets at unprecedented scale. Follow-up works include BASIC [67], LiT [93], BLIP [44], SLIP [57], *etc*. Without loss of generality, this study takes a special interest in CLIP.

**Image-text datasets.** Web-crawled image captioning data are typically formatted as image-text pairs, which can be crawled from the web. Existing works provide a wide range of options across

scales, including MS-COCO [14], CC-3M [76] and 12M [12], YFCC-100M [80] and 15M [68], WIT [79], SBU [62], RedCaps [20], LAION-400M [73] and 2B/5B [74], and MetaCLIP [91], *etc*. This work considers those with both metadata and pre-trained CLIP publicly available, *i.e.*, CC-12M, YFCC-15M, LAION-400M/2B, and MetaCLIP-400M/2.5B.

### B.2   Obtaining class frequency statistics

This study specifically examines the classes of ImageNet [18], which encompasses 1K common object categories. To obtain the class distribution on image-text datasets, we follow the common practice [27, 77, 91] to query captions with class names and their WordNet [56] synset. In implementation, we also loosen the sub-string matching condition to set-level matching (overlooking the order of words) for a higher recall, and manually introduced negative words (*e.g.*, 'vehicle', 'truck' for class 'ram', 'bird', and 'wing' for class 'crane') to reduce false positives. Besides, we normalize letters to lowercase, remove non-letter and non-number symbols, and lemmatize words to nouns. For MetaCLIP, which provides a readily available distribution of 500K concepts, we simply summed up the statistics of target concepts (classes). And for other datasets, we ran the search over all captioning data.

### B.3   Open-source CLIP models

The models are collected from the models of OpenCLIP [15]. We select models that have captions or metadata of the pre-training dataset publically available, and restrict the backbones to ResNet [33], ConvNeXt [52], and ViT [22]. The remaining set comprises 41 models covering different model architectures (6 ResNets, 8 ConvNeXts, and 27 ViTs), model scales (ResNet-50/101, ConvNeXt-B/L/XL, and ViT-B/L/H/G), data scales (from 12M to 2.5B), training schedules, and optimization techniques. An overview of the results of these models is provided in Fig. 18.

## C   Details about the controlled study

### C.1   Training details

Our training settings follow the common practice in [27], CLIP experiments utilize cross-entropy losses and the AdamW optimizer. The initial learning rate is set to 0.001, and a cosine-annealing learning rate schedule with 500 warmup steps is employed. The hyper-parameters for AdamW are $\beta_1 = 0.9$, $\beta_2 = 0.999$, and $\epsilon = 10^{-8}$. The batch size is set to 1024. Model training lasts for 32 epochs. We also tried training 90 epochs to match that of SL but found 32 epochs is enough for convergence and longer training has no notable benefit.

SL models are trained using SGD with Nesterov momentum for 90 epochs. The weight decay is set to 0.0001, momentum to 0.9, and batch size to 256. The initial learning rate is 0.1 and is decayed by 0.1 at epochs 30, 50, and 70.

To maximally align details with CLIP, both methods adopt the slightly modified ResNet structure as in [68]. The augmentation pipeline is also kept consistent: random resized crop to size 224 with a scale range of $(0.9, 1.0)$, followed by normalization with a mean of $(0.48145466, 0.4578275, 0.40821073)$, and a standard deviation of $(0.26862954, 0.26130258, 0.27577711)$. Note that this data augmentation pipeline is notably weaker than those commonly used by SL.

### C.2   Details about text formation in ImageNet-Captions

For ● template-based captions, the caption of an image is generated using a randomly sampled template from 80 class templates provided in [68], *e.g.*, "a photo of a [class]". If synsets are used, the class name [class] is also randomly sampled from its synsets. For ● natural-language captions, we refer to Fig. 2 (upper) for an example image and corresponding text metadata (including title, description, and tags). More examples can be found in Fig. 3 of [27]. The way captions are created is simply by concatenating metadata together with spacing. *E.g.*, if the [title] is "A phone call and night", and the [description] is "I might have a thing with telephones. . . ", then the resulting caption is [title description]: "A phone call and night I might have a thing with telephones. . . ". This follows the practice of [27], and is also the way CLIP [68] curates caption data from YFCC-15M.

## C.3 Details about vocabulary subsampling in SL

The training vocabulary refers to the label set that a model classifies at a specific training iteration. Given a mini-batch of samples, a minimal label set is formed as the union of all GTs in this mini-batch. If the expected vocabulary size is not met, we additionally sample classes from the remaining, and the probability a class is selected is determined by the *frequency* of the corresponding class in the pre-training data. Note that the sampling is performed at the class level, which differs from the sampling strategies in long-tail learning that are done at the sample level. We also tried *uniform* sampling, *i.e.*, treating each class with equal probability, which yielded slightly weaker results.

**Discussions.** For SL, vocabulary subsampling refers to randomly reducing the size of candidate classes (akin to dropout on the classification head) when classifying an image during training. 1) Regarding how it works, Fig. 3a ($y$-axis) shows it effectively reduces the model's predictions' correlation to class frequencies, a key indicator of classifier's bias. 2) Regarding why this technique can de-bias classifiers, our intuition is that this plays a similar role to dropout: the classifier is regularized to put equal importance on all classes. Biases still exist in the subsampled classes, but the gradients cancel out each other during training. 3) Regarding why frequency-based sampling works better than dropping all classes with equal probability, we hypothesize that the dropping operation can de-bias the classifier regardless of how classes are selected, and sampling by frequency is more helpful for representation learning. The intuition comes from the finding in long-tail learning that resampling data by inverse frequency helps de-bias classifier, but harms representation learning [38, 97].

## C.4 Details about models' heads

For CLIP experiments, the text encoder is trained from scratch by default. If the text encoder uses frozen CLIP, this means the text encoder is initialized by the pre-trained CLIP weights from [68]. During training, the parameters of the text encoder remain unchanged. In the CLIP init setting, after initialization, the text encoder is also fine-tuned in the training process. Further, for RoBERTa experiments, we follow the implementation of [15] and replace the text encoder with pre-trained RoBERTa [51] available on HuggingFace [89]. This is kept frozen during training, as we found fine-tuning it results in NaN loss.

For SL experiments, we replace the commonly used linear classifier with a prototypical classifier to better follow CLIP's structure. This means the bias term in this linear layer is omitted, and both the feature from the backbone and the classifier's weight are $\ell_2$-normalized, thus weights in the linear layer can be viewed as a set of prototypes (feature vectors). To facilitate optimization, a learnable scaler with a maximum scale of 100 is added as CLIP [68] to upscale logits. For the setting using fixed prototypes obtained from CLIP, we format each class to a sentence using the template "a `[class]`", feed them to the text encoder of a pre-trained CLIP, and keep the output class-wise text features as SL model's classification head/prototypes.

## C.5 Details about image-text dataset variants

**ImageNet-Captions subsets.** Starting from the original ImageNet-Captions [27], we take only image-text pairs that correspond to the 100 classes of Tian et al. [81], thus obtaining a 100-class subset called ImageNet-Captions-100. Besides, we randomly sample from ImageNet-Captions and construct a subset that is of the same scale as ImageNet-Captions-100 but with the same number of classes as ImageNet-Captions. This subset is called ImageNet-Captions (10%). Note that it is of the same scale of ImageNet-Captions-100, and not necessarily 10% of ImageNet-Captions.

**LAIONet variants.** LAIONet [77] is a subset of LAION-400M [73] created by matching between ImageNet class synsets and captions. Items with low CLIP text similarity between the caption and class definition are filtered out to reduce label noise. Our reproduction sets 0.7 as the default threshold, and 3.26M images are successfully crawled. Experiments in Sec. 3.4 consider LAIONet variants filtered with different text-definition similarity thresholds: 0.7, 0.75, 0.8, 0.82, and the sizes of corresponding LAION-400M subsets are originally 3.26M, 1.93M,

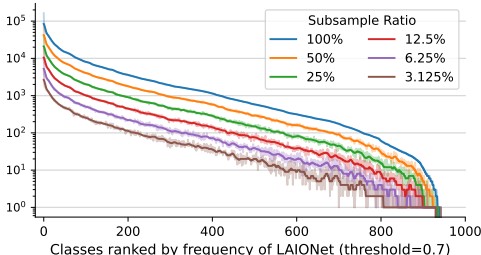

Figure 16: Distribution of LAIONet subsets.

0.88M, and 0.58M. We then randomly subsample them to be the same scale as ImageNet-Captions (0.45M). Besides, the variant that matches the class distribution of ImageNet-Captions is sampled from the 3.26M version, and the scale is also kept the same as ImageNet-Captions. In addition, experiments in Sec. 3.5 use LAIONet subsets randomly sampled from the 3.26M version (threshold set to 0.7), at the portion of $1/1$, $1/2$, $1/4$, $1/8$, $1/16$, and $1/32$, respectively. The distributions of these randomly sampled subsets are shown in Fig. 16.

**CC-12M-Cls and YFCC-15M-Cls.** These are classification subsets of CC-12M and YFCC-15M that have corresponding class labels of 1K ImageNet classes for each image. The curation process follows Fang et al. [27], except that we allow class name matches to be not case-sensitive. In comparison to LAIONet, it is simply substring matching without filtering. The resulting datasets are at a scale of 3.48M (CC-12M-Cls) and 2.90M (YFCC-15M-Cls), respectively.

### C.6    Evaluation setting

Unless otherwise specified, the evaluation of models is all performed on ImageNet validation split. For CLIP, the default zero-shot classification setting is applied. That is, each class is embedded as an average vector of text features produced using 80 class templates provided in [68]. Then for both CLIP and SL, the predicted class is that of the nearest neighbor class prototype.

### C.7    Computing resources

Experiments are conducted on NVIDIA A100 GPUs. Each CLIP and SL training experiment runs on 4 GPUs in parallel, and there are roughly 400 experiments (data points) for the controlled study.

## D    Details about DINO experiments

### D.1    Preliminaries

**Self-supervised learning from pseudo-labels.** It is natural to extend SL to self-supervised settings for representation learning, as long as pseudo-labels are available. Earlier work [8] applies $k$-means clustering to deep features and takes cluster assignments as pseudo-labels. Following works [2, 10] reform pseudo-labeling as optimal transport and solve it with the Sinkhorn Knopp algorithm. This is then simplified by DINO [11] with centering and sharpening operations on the model's predictions, and extended to soft labels (thus called self-distillation instead of self-labeling).

**Knowledge DIstillation with NO labels (DINO).** DINO [11] is a discriminative self-supervised visual pre-training method. The pretext task is formulated as self-distillation: enforcing the student model's predictions to be close to teacher models' soft pseudo labels. The input to two models are random augmented views of the same image, and the teacher model is updated as the exponential moving average of the student model (also called "mean teacher"). DINO learns a set of prototypes (feature vectors) as the classification head, and is used by student and teacher models to produce logits and pseudo labels. Since the prototypes resemble a classification head, the aforementioned vocabulary subsampling strategy can also be similarly applied to DINO.

### D.2    Training details

The training details follow the suggested practices of DINO [11] for training ResNets. That is, train using SGD optimizer with a base learning rate of 0.03, and fixed weight decay of 0.0001. The scale of global crops is $(0.14, 1)$, and the scale of local crops is $(0.05, 0.14)$. Other hyper-parameters are kept as default. We use the ResNet-50 backbone with the same structure as Radford et al. [68], and train for 100 epochs with a batch size of 1024.

The last layer of DINO's projection head is equivalent to a set of prototypes, thus it is natural to integrate the techniques experimented to be valid on classification models. We keep the total number of prototypes to 65536 as default.

For vocabulary sampling-based DINO, we subsample the same set of prototypes for the teacher and student models and compute the self-distillation loss on this restricted prototype set. The vocabulary (prototype set) is shared in a mini-batch, and different across training iterations.

### D.3 Transfer learning details

**Datasets and metrics.** We test models' transfer learning performance on the benchmark initially proposed in [40], and adopt the implementation from [23]. The datasets in this benchmark include: Food-101 [7], CIFAR10/100 [42], Birdsnap [5], SUN397 [90], Stanford Cars [41], FGVC Aircraft [54], PASCAL VOC 2007 [24], Describable Textures (DTD) [16], Oxford-IIIT Pets [65], Caltech-101 [28], and Flowers-102 [59]. The evaluation metric is mostly top-1 accuracy, with exceptions of mean per-class accuracy on FGVC Aircraft, Oxford-IIIT Pets, Caltech-101, and Flowers-102, and 11-point mAP on PASCAL VOC 2007.

**Linear probing.** Image features are extracted from the backbone of the teacher model following [11]. Then following [23], we train an $\ell_2$-regularized multinomial logistic regression classifier on frozen features extracted from the backbone. The model is optimized using L-BFGS on the softmax cross-entropy objective. No data augmentation is applied, and the images are resized to 224 pixels along the short size using bicubic resampling and center-cropped to $224 \times 224$. The hyper-parameters for $\ell_2$-regularization are searched from 45 logarithmically spaced values between $10^{-6}$ and $10^5$.

## E  Extended results

### E.1 Examples of class distribution and CLIP performance

In Fig. 17, we provide an example of the distribution of subsampled classes and per-class zero-shot accuracy of CLIP (ViT-B/32) pre-trained on ● LAION-400M and ■ MetaCLIP-400M accordingly. The head classes are easy to be found the web, *e.g.*, "T-shirt", "mobile phone", "throne", and "goose", *etc*. In contrast, the tail classes are dominated by fine-trained biological concepts, ranging from "barn spider", "earth star fungus", to "gyromitra". Collecting such data is hard and requires expert knowledge. Despite this, we find both models can achieve good performance on some tail classes.

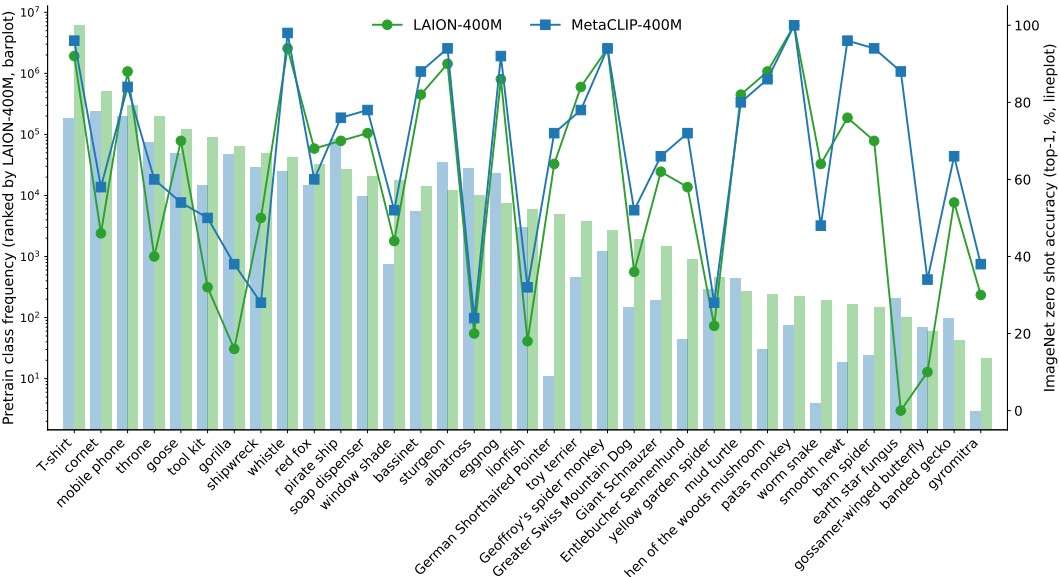

Figure 17: Examples of the distribution of subsampled classes (bar plot), and per-class zero-shot accuracy (line plot) of CLIP (ViT-B/32) pre-trained accordingly (● LAION-400M and ■ MetaCLIP-400M). Both models show a weak correlation between class frequency and accuracy.

### E.2 Extension of Fig. 1b with per-model results

In supplement to the analysis in Fig. 1b where results of CLIP are averaged by the dataset it trains on, we provided more detailed results of CLIP in Fig. 18. Besides zero-shot classification results on ImageNet [18], Fig. 18 also provides results evaluated on ImageNetV2 [70]. Results are consistent.

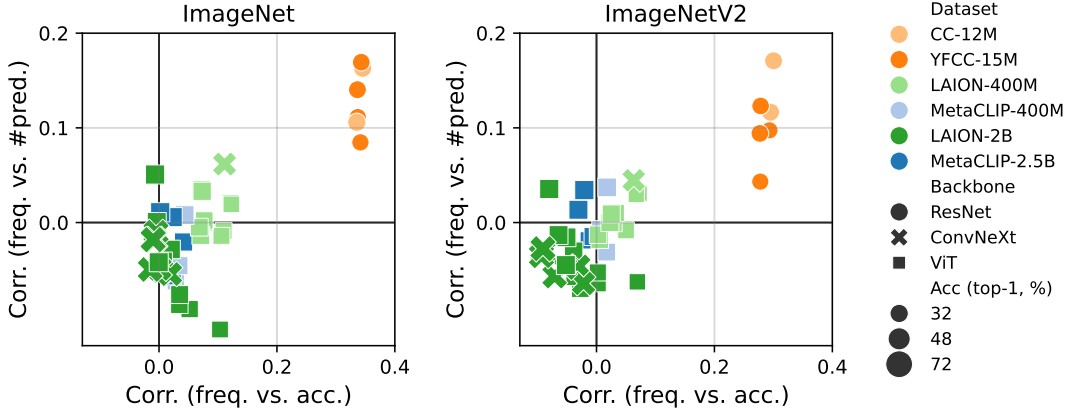

Figure 18: An overview of the correlation between open-source CLIP models' per-class accuracy, and prediction distribution with pre-training data's class frequency. The weak correlation to sample frequency is consistent whether evaluated on ImageNet [18] or ImageNetV2 [70].

### E.3 Extension of Fig. 3 with language pre-training

In supplementary to the analysis in Fig. 3, which is conducted under the setting that models are trained from scratch. Here we also provide the results that all models are trained using frozen CLIP text encoders/heads in Fig. 19. We find that the results are generally consistent with those in the main paper. In addition, we find language pre-training provides a shortcut to models and allows them to leverage language supervision (CLIP) and debiased pretext tasks (SL) with higher effectiveness. This is supported by the sharper slopes in (a, blue line) and (b, green line) in comparison to Fig. 3.

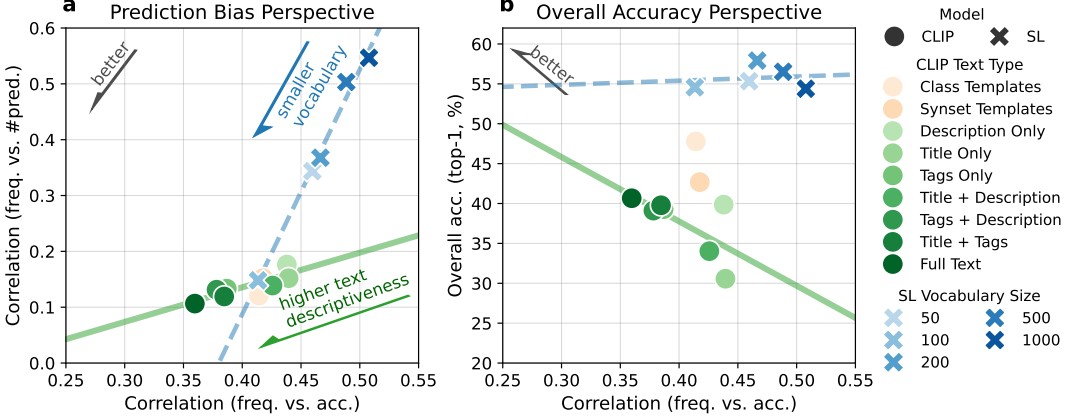

Figure 19: Results on IN-Caps about caption diversity and vocabulary size. Both CLIP and SL use frozen text encoders/prototypes from *pre-trained* CLIP. The trends are mostly consistent with Fig. 3. In addition, the models using ● template-based supervision are (a) less biased and (b) show better accuracy than the training-from-scratch counterparts in Fig. 3, indicating the knowledge in language pre-training to be obtained by CLIP. This also holds true for SL and ● natural language-supervised CLIP, as supported by shaper slopes in (a, blue line) and (b, green line).

### E.4 Extended visualizations of CLIP's multi-modal feature space

In supplement of Fig. 7b, we also plot the vision feature centers and corresponding sample features of some classes in Fig. 20. Results are produced by a CLIP ViT-B/32 model pre-trained on LAION-400M, and obtained by inferencing on the ImageNet validation split. Note that vision and text features are plotted separately due to the modality gap (despite being in the same feature space) [47]. Fig. 20a shows the features of images from some subsampled classes, and corresponding vision feature centers.

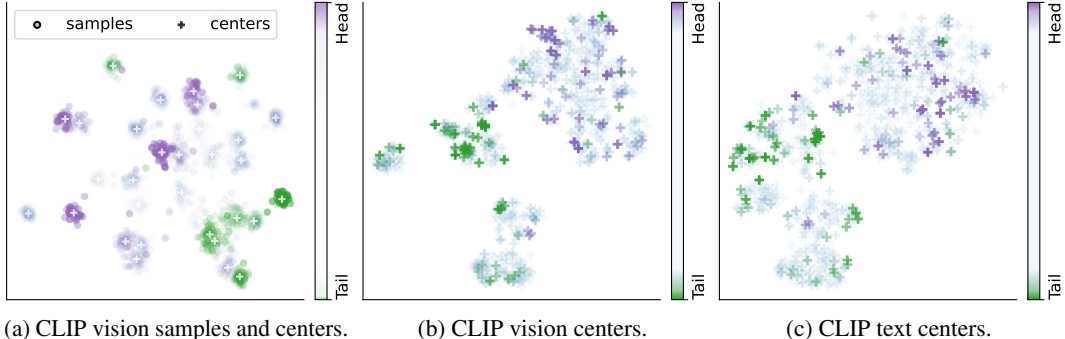

(a) CLIP vision samples and centers.  (b) CLIP vision centers.  (c) CLIP text centers.

Figure 20: t-SNE visualization of samples encoded by CLIP vision/text encoders in the multi-modal feature space (on ImageNet validation set). (a) Images encoded by CLIP vision encoder, and their class-wise mean features. Classes are subsampled. (b) Vision feature centers of all ImageNet classes. (c) Class templates encoded by CLIP text encoder, the same as Fig. 7b. Vision and text features are plotted separately due to the modality gap (despite being in the same feature space) [47].

In coherence to results in Fig. 7a.2, there is not a clear tendency on whether head or tail classes form compactor clusters. In addition, Fig. 20b and Fig. 20c show the vision and text feature centers of all ImageNet-1K classes, where head and tail classes are highlighted. The vision feature centers are produced by averaging sample features by classes, and the text feature centers are as of the classifier used by CLIP, as described in Appx. C.6. The margins between tail classes encoded by the vision encoder are notably smaller. In contrast, tail class centers produced by the text encoder are better separated. This phenomenon might be connected with the modality gap [47], and is of research value for future explorations.

### E.5 Original numeric data of DINO transfer learning results

In Tab. 2, we provide the original numeric data used to obtain Fig. 10 for reference.

Table 2: Linear probing evaluation results of DINO variants pre-trained on LAIONet for 100 epochs. Extreme data imbalance makes LAIONet much harder for DINO to learn transferrable representations, and vocabulary subsampling strategy effectively helps DINO overcome such defects.

| Dataset | \|Voc\| | Aircr | Birds | C101 | Cars | CF10 | CF100 | DTD | Flower | Food | Pets | SUN | VOC | Avg |
|---|---|---|---|---|---|---|---|---|---|---|---|---|---|---|
| *Results of vanilla DINO* | | | | | | | | | | | | | | |
| ImageNet | 65536 | 27.0 | **37.1** | **82.3** | 23.6 | **86.4** | 62.9 | 68.7 | **80.8** | **55.8** | **66.4** | 57.0 | **81.6** | **60.8** |
| LAIONet | 16384 | 29.7 | 28.2 | 78.2 | 22.5 | 83.6 | 60.7 | 67.9 | 78.3 | 49.5 | 55.0 | 55.9 | 76.1 | 57.1 |
| LAIONet | 65536 | 26.7 | 24.6 | 77.8 | 24.6 | 83.5 | 60.0 | 67.1 | 79.3 | 48.6 | 55.5 | 55.4 | 77.3 | 56.7 |
| *Results of DINO + vocabulary sampling (65536 prototypes in total)* | | | | | | | | | | | | | | |
| LAIONet | 1024 | 30.8 | 27.2 | 78.6 | 23.8 | 83.9 | 61.4 | 68.1 | 80.5 | 50.7 | 57.7 | 56.0 | 77.2 | 58.0 |
| LAIONet | 4096 | 30.3 | 30.1 | 78.9 | 24.8 | 84.6 | 63.4 | 69.5 | 77.7 | 53.3 | 61.0 | 56.9 | 78.6 | 59.1 |
| LAIONet | 16384 | **32.2** | 31.2 | 79.4 | **25.2** | 85.4 | **63.9** | **70.2** | 79.1 | 54.3 | 62.2 | **57.7** | 79.0 | 60.0 |

### E.6 Zooming in at the class distributions (linear scale)

To provide a clearer image of the imbalanced class distribution of pre-training datasets, we show a zoomed-in version of Fig. 1a with linear scale in Fig. 21. Also, we see that MetaCLIP does successfully alleviate the dominance of head classes. But note that unfortunately, all datasets are still extremely imbalanced, and how to improve models' robustness to it is still to be explored.

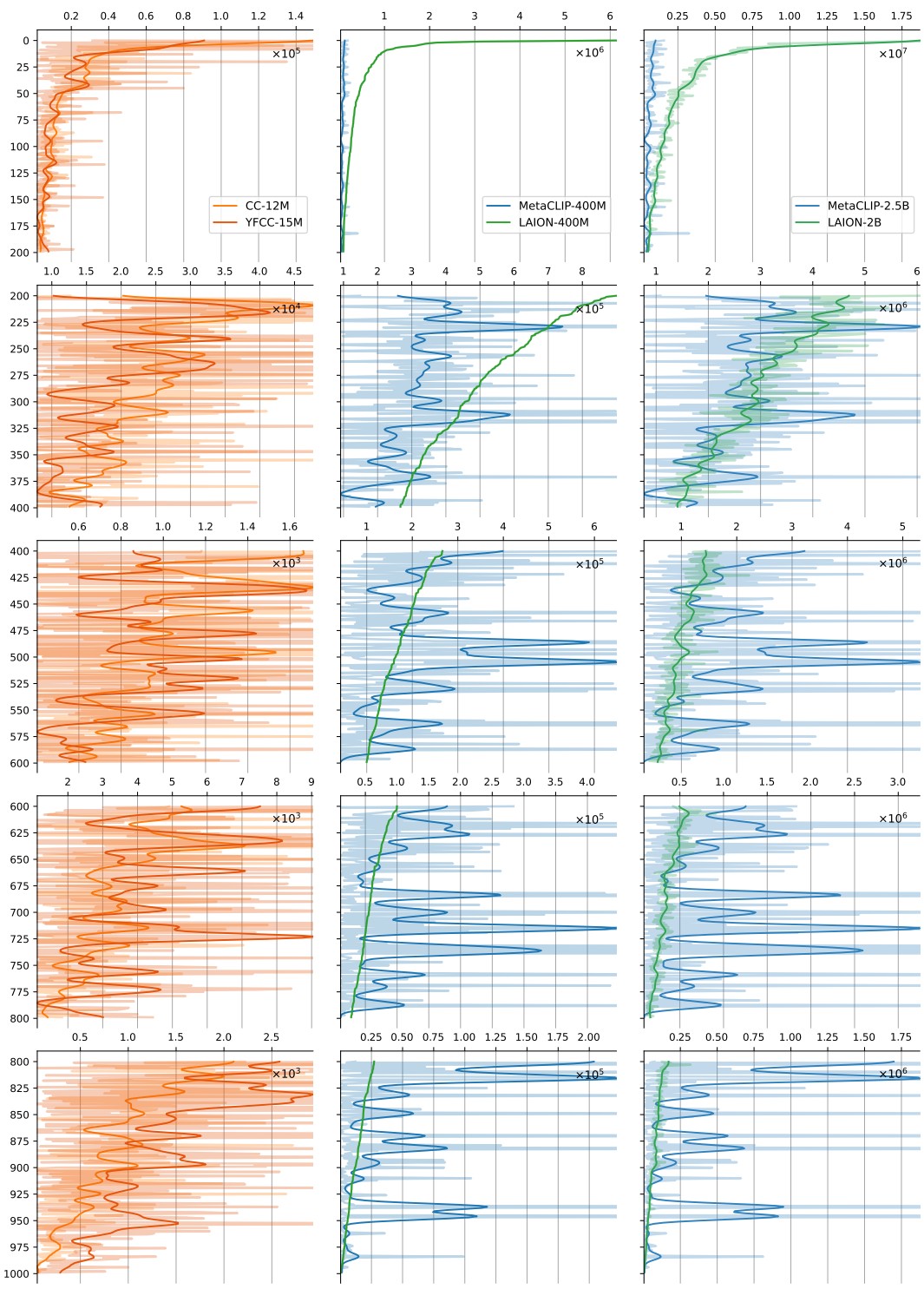

Figure 21: A zoom-in version of Fig. 1a showing class frequencies (linear scale) ranked by LAION-400M. An imbalanced class distribution is shared across datasets.

