# OpenReview forum: "What Makes CLIP More Robust to Long-Tailed Pre-Training Data? A Controlled Study for Transferable Insights"
_NeurIPS.cc/2024/Conference — NeurIPS 2024 poster_

### Official Review · Reviewer_WXxS · 2024-06-23

**Soundness:** 3
**Presentation:** 3
**Contribution:** 2
**Rating:** 5
**Confidence:** 3

**Summary:**

This author empirically studies the mechanisms behind CLIP's generalization ability from multiple aspects. Based on the findings, they further transfer the merits of CLIP to other scenarios like supervised learning and self-supervised learning.

**Strengths:**

1. The proposed problem is worth exploring.
2. The experiments are comprehensive.

**Weaknesses:**

1. The authors studied the robustness/generalization of CLIP from various points, but some points are known to the community. Sections 3.2 and 3.4 claim that higher caption density and larger data volumes in training data contribute to robustness in terms of imbalances. However, this observation appears to be well-understood in the community. It is generally accepted that high-quality, richly described data is preferable.
2. I appreciate the presented comprehensive studies. However, the settings and conclusions in Section 3 are occasionally unclear. I recommend adding a brief conclusion to each subsection to enhance clarity and coherence.
3. In Section 4.2, which examines subsampling vocabulary in self-supervised learning with DINO, the impact of this technique is only demonstrated on models with a default dimension of 65,536. It remains unclear if this approach would be equally effective on other self-supervised learning models that utilize significantly smaller default dimensions, such as 2,048.

**Questions:**

1. The methodology for handling training data during the random subsampling of vocabulary, as discussed in Section 3.3, remains ambiguous. With a batch size of 1024 and a reduced vocabulary size of 500, how are the remaining samples, whose labels do not fit within the reduced vocabulary, treated? This clarification is essential for understanding the implications of the approach on the learning process.
2. Is there a correlation between smaller vocabulary sizes and smaller batch sizes in these experiments?
3. The rationale behind the selection of specific thresholds (0.7, 0.75, 0.8, 0.82) in Figure 5 is unclear, particularly the choice of 0.82 over 0.85.

**Limitations:**

1. No code is provided at this stage.

---

> ### Author Rebuttal · Authors · 2024-08-06
>
> Dear reviewer WXxS, thanks for helping improve our paper, and your concerns are responded to as follows:
>
> > L1: No code is provided at this stage
>
> We have organized the code and provided it to the AC with a separate comment according to the rebuttal policy, please contact the AC for access. The code contains all code and data ranging from concept frequency estimation, dataset curation, and training and evaluating SL/CLIP/DINO models and their variants considered. We also provided notebooks to replicate all figures present in the paper. We will make it public to help future research.
>
> > W1: It is generally accepted that high-quality, richly described data is preferable
>
> 1. We agree, but we also would like to note that we are the first to study the effect of these factors in terms of robustness to *imbalanced pre-training data*, contrasting existing works that focus on *OOD evaluation data*. We believe this viewpoint is timely given the current research trend that lacks a detailed study of factors in the pre-training data and how they interact with model factors.
> 2. Please note that not all data factors we studied could enhance robust performance. Eg, we also find that more severe data imbalance in web data can harm robustness (Fig. 5).
> 3. We also note that the benefits of larger-scale data do not come for free, it also requires a suitable model. Eg, CLIP’s robustness consistently improves as data scales up, while not for traditional SL, and even not for modern generative models (LLM & MLLM, see our response to reviewer tULz’s Q1).
>
> > W2: Settings and conclusions in Sec. 3 are occasionally unclear
>
> Thanks for pointing this out. We put most details of the setting in the appendix due to the page limit. In the next version, we’ll incorporate more details in the main text and highlight the takeaway messages according to your suggestion.
>
> > W3: It remains unclear whether subsampling vocabulary helps other SSL methods
>
> 1. We choose DINO because its pretext task involves pseudo-labeling with online-learned prototypes – each standing for a virtual class (called “last_layer” in the code). Its feature dimension is 2048 and the projector dimension is 256 by default, similar to other SSL methods. Other contrastive methods discriminate between randomly sampled images in training, and vocabulary subsampling is not applicable.
> 2. We recently noticed a study [a] that shows the learnable prototypes of DINO (akin to the classifier of SL) may be biased to imbalanced data and many of them collapse. Our strategy might help alleviate such collapse and encourage a more balanced utilization of these prototypes.
> 3. We agree this finding is valuable for extension to other SSL methods, especially clustering-based SSL, including SwAV, DINO, iBOT, DINOv2, etc.
> 4. We also would like to emphasize that our experiments of vocabulary subsampling are intended to verify the underlying rebalancing/debiasing mechanism of CLIP can be transferred to other domains (SL and SSL). We do not intend to propose a novel method, but to point out problems to solve (model-side debiasing is widely needed when training on web data). We call for, and it is very likely better methods for debiasing large-scale trained SSL/LLM/MLLM to be developed in the future.
>
> Reference:
>
> [a] Govindarajan et al., On Partial Prototype Collapse in Clustering-Based Self-Supervised Learning
>
> > Q1: When applying vocabulary subsampling, how are remaining samples whose labels do not fit treated
>
> We ensure all GT labels in a batch are covered after subsampling. Please check Sec. C.3 for further details.
>
> > Q2: Is there a correlation between smaller vocabulary sizes and smaller batch sizes
>
> Our experiments kept the batch size unchanged. However, we do expect the optimal vocabulary to be a smaller value correlating with the batch size, thus rebalancing learning signals while forming meaningful classification tasks.
>
> > Q3: How are the thresholds in Fig. 5 selected
>
> The choice is to ensure we can obtain images as much as the scale of ImageNet-Captions (0.45M). When using thresholds higher than 0.82, the remaining number of images is small, eg, 0.26M for threshold 0.85.

---

> > ### Author Response · Authors · 2024-08-13
> > **Looking forward to having further discussions**
> >
> > Dear Reviewer WXxS,
> >
> > Thank you so much for your time and efforts in reviewing our paper. Hope our rebuttal has addressed your concerns, and we will revise our paper based on your feedback. We are happy to discuss with you further if you have other concerns. Thanks for your time!
> >
> > Best regards,
> >
> > Paper 4735 authors

---

> > > ### Comment · Area_Chair_X3xU · 2024-08-13
> > >
> > > Reviewer WXxS, do you have any additional questions or feedback?

---

> ### Author Response · Authors · 2024-08-14
> **A TL;DR version of our response**
>
> Dear Reviewer WXxS,
>
> We sincerely appreciate the considerable time and effort you have dedicated to reviewing our paper. Your thought-provoking questions and insightful feedback have been instrumental in improving the quality of our work.
>
> In response to your queries and concerns, we have carefully crafted detailed answers. We are keenly aware of the numerous submissions you are managing and deeply appreciate your attention to our research. Given the time constraints during this busy discussion period, we have summarized the main points of our rebuttal below. **This concise overview aims to provide a quick reference while respecting your valuable time.**
>
> - Code with all implementation details is provided to the AC and will be publicly available very soon (L1).
> - We study CLIP's robustness to **imbalanced pre-training data**, while prior works study robustness to **OOD evaluation data**, and prior conclusions do not directly apply to our setting; We also find data factors that harm robustness; We also identify important model factors that make CLIP stand out, and **even modern LLM/MLLMs trained with large-scale data suffer from imbalanced data** (W1).
> - Vocabulary subsampling could be a general way to rebalance (pseudo-)labels in supervised and unsupervised settings. Still, in this section we intend to pose a general problem (**model-side debiasing is widely needed when training on web data**) rather than proposing novel solutions (W3).
> - Details of settings are provided in the appendix and questions are explained in our response (W2 & Q1/2/3).
>
> We sincerely hope that our responses have satisfactorily addressed your concerns. We remain open and enthusiastic about addressing any additional questions you may have, as your continued input would greatly contribute to further improving our work.

---

### Official Review · Reviewer_iwVX · 2024-07-11

**Soundness:** 3
**Presentation:** 3
**Contribution:** 2
**Rating:** 5
**Confidence:** 3

**Summary:**

This paper explores the remarkable robustness of CLIP training compared to supervised training when dealing with real-world imbalanced datasets. This advantage can be attributed to dynamic classification using a subset of classes. Furthermore, the paper shows that robustness and discriminability improve with denser language supervision, larger data scales, and broader open-world concepts. The experiments are mainly conducted on ImageNet-captions and LAIONet to validate the main claims.

**Strengths:**

- This paper reveals the underlying distribution (using ImageNet concepts) of several widely-used vision-language pretraining datasets.
- The issue of vision-language learning under imbalanced data distribution is critical.
- This paper presents the finding that CLIP training is more robust than supervised learning and explains the underlying mechanism behind this phenomenon.

**Weaknesses:**

-	This paper explores vision-language pretraining (VLP) under imbalanced distributions primarily from a data perspective, focusing on language supervision, data scaling, and open-world concepts. These factors, known to enhance robust performance, have been discussed in several previous studies.
-	Other factors beyond data may also affect the robustness of VLP under imbalanced distributions. For instance, from a loss perspective, comparing the learning abilities of CLIP loss and cross-entropy loss in supervised settings, particularly under imbalanced conditions, needs discussion.
-	In the language supervision section, a more comprehensive experiment would compare self-supervised contrastive learning methods, as they share the same contrastive objective. This comparison would isolate the differences between contrastive and cross-entropy losses, focusing solely on variations in language supervision.
-   It seems that the concept frequency estimation in this paper may suffers from noise, multi-label structure.
-   The evaluation is mainly focusing on classification.

**Questions:**

please refer to the weakness part.

**Limitations:**

The limitations are discussed in this paper. No potential negative social impact.

---

> ### Author Rebuttal · Authors · 2024-08-06
>
> Dear reviewer iwVX, your questions are answered below:
>
> > W1: The factors studied are primarily from a data perspective and known to enhance robust performance
>
> 1. To the best of our knowledge, most existing works study CLIP’s robustness to OOD evaluation data, while we are the first to study CLIP’s robustness to *imbalance pre-training* data from these perspectives. We believe this viewpoint is timely given the current research trend of brute-force scaling up, without a detailed study of factors in the pre-training data and how they interact with model factors.
> 2. Please note that not all data factors we studied could enhance robust performance. Eg, we also find that more severe data imbalance in web data can harm robustness (Fig. 5).
> 3. We also note that the benefits of larger-scale data do not come for free, it also requires a suitable model. Eg, CLIP’s robustness consistently improves as data scales up, while not for traditional SL, and even not for modern generative models (LLM & MLLM, see our response to reviewer tULz’s Q1).
>
> > W2: Other factors beyond data should be considered, eg, from a loss perspective
>
> Actually, this is exactly what motivated us to identify the factor “dynamic classification as pretext task” (Sec. 3.3). We tried to align every detail between SL and CLIP, and finally found applying CLIP loss to SL to be a detrimental factor. We then dug into the mechanisms of CLIP loss, and found CE+vocabulary subsampling to have the same effect.
>
> In addition, we also experimented on factors like the architecture of the vision backbone and text backbone, vision pre-training, stronger data augmentation, larger batch size, and test-time prompts, and did not find noticeable effects. These factors are thus not presented in the paper. A discussion on the motivation behind the factors we considered is provided in Sec. A.6.
>
> > W3: Self-supervised learning methods should also be discussed to show the effect of language supervision
>
> Thanks for raising this concern. We additionally trained a self-supervised learning method DINO on ImageNet-Captions, and used k-NN to evaluate its classification performance. In comparison to SL and CLIP, it further verifies the effectiveness of language supervision.
>
> | Model                    | SL   | DINO (k-NN) | CLIP |
> | ------------------------ | :--: | :---------: | :--: |
> | Corr. (freq. vs. acc.)   | 0.52 | 0.35        | 0.28 |
> | Corr. (freq. vs. #pred.) | 0.57 | 0.20        | 0.08 |
>
> > W4: It seems the concept frequency estimation may suffer from noise, multi-label structure
>
> 1. This is why we tried to study as many factors on IN-Caps as possible. However, it is still expected that we verify these factors on other datasets to better support our findings (as requested by Reviewer XjUF), and provide a complete discussion on factors not supported by IN-Caps (eg, data distribution and data scale), thus facing an inevitable trade-off. Accurate concept frequency estimation of web datasets is an emerging topic and we believe it is valuable for future research.
> 2. Regarding LAIONet, the authors are aware of the label noise and explored applying a filter. They empirically found this to be effective in overcoming such noise (see Fig. 11 in LAIONet’s arXiv preprint). When replicating LAIONet, we are conservative on the choice of the filtering threshold, trying to make the selected images have highly confident labels.
> 3. We also explored an even simpler strategy: set-based substring matching (detailed in Sec. B.2). We applied this strategy to estimate the frequency in large-scale datasets (eg, LAION-400M). In comparison to advanced estimation techniques using ChatGPT and Llama2 (Parashar et al.), our method shows very consistent results (correlation of 0.8-0.9, see Sec. A.4 for details).
>
> > W5: The evaluation is mainly focusing on classification
>
> Please note that we study the effects of data imbalance in pre-training data, which raises challenges in accurate estimation of labels, control of ablating factors, and high training cost of each experiment (1-2 days on 4xA100 GPUs each experiment, more than 400 experiments in total). Evaluations on classification are a result of trading off these factors and presenting as comprehensive an analysis as possible.

---

> > ### Author Response · Authors · 2024-08-13
> > **Looking forward to having further discussions**
> >
> > Dear Reviewer iwVX,
> >
> > Thank you so much for your time and efforts in reviewing our paper. Hope our rebuttal has addressed your concerns, and we will revise our paper based on your feedback. We are happy to discuss with you further if you have other concerns. Thanks for your time!
> >
> > Best regards,
> >
> > Paper 4735 authors

---

> > > ### Comment · Area_Chair_X3xU · 2024-08-13
> > >
> > > Reviewer iwVX, do you have any additional questions or feedback?

---

> > > ### Comment · Reviewer_iwVX · 2024-08-14
> > >
> > > Thanks for the detailed responses. Most of my concerns have been addressed. I appreciate the additional experiments w.r.t self-supervised learning method DINO on ImageNet-Captions. Overall, I notice the importance of this paper on understanding imbalanced multimodal learning, and would like to raise my score to '5'.

---

> > > > ### Author Response · Authors · 2024-08-14
> > > > **Thank you!**
> > > >
> > > > Dear Reviewer iwVX,
> > > >
> > > > We appreciate your positive feedback and are truly delighted to see that our response has addressed your concerns and questions.
> > > >
> > > > Thank you once again for dedicating your time and effort to reviewing our work and providing us with insightful suggestions!

---

### Official Review · Reviewer_XjUF · 2024-07-13

**Soundness:** 3
**Presentation:** 3
**Contribution:** 3
**Rating:** 4
**Confidence:** 4

**Summary:**

Motivated by the novel observation on CLIP's robustness against data imbalance, this paper provides extensive analysis on factors that determine such robustness, including language supervision, classification templates, data scaling, distribution, and open-world concepts. The paper shows that such factors are transferrable to supervised or self-supervised learning to reach CLIP-level robustness.

**Strengths:**

- It is novel observation that CLIP shows robustness against data imbalance.
- The factors, including data scale and dynamic classification templates, that determine the robustness have practical insights in data imbalance problems.

**Weaknesses:**

1. The importance of data scale predominates other factors:
- in ablation studies for data scaling, the setup with frozen CLIP text encoder header only shows the reasonably low correlations (still, much higher than large-scale examples shown in Fig.1) among other setups. The similar results are observed in the ablation studies in different distributions (Fig. 5-b) as well as extreme case (Fig. 8), where given the same distribution, the setup with frozen CLIP text encoder shows superior correlations. Are these results indicating that the large-scale data pretraining predominates the CLIP's robustness on data imbalance? In other words, given large-scale data, e.g., metaCLIP-2.5B, how ineffective are those factors studied in the paper? Are they still influential or negligible?
- In Figure 5, I find it difficult to clearly understand regarding distribution shifts. It needs more details regarding the distribution shifts and the meaning of the dashed line. How are the results calculated and what is the take-away message?
- In Figure 6, the CLIP with YFCC-15M shows stronger correlations than that with YFCC-15M-cls, which cannot be explained by the authors’ claim that open-world datasets improve robustness.

2. In section 4.1, additional results on SL using init head + sub voc. are necessary to fairly ablate the factors for showing transferability to SL.

3. It is difficult to clearly understand the setup in Section 4.2. It lacks detailed explanation on setups for comparison, and the measures only include downstream task performance, not the correlations.

**Questions:**

- How did you calculate the ImageNet-class frequency in web-scale datasets in Fig.1 (a)?
- Regarding weakness 1., experiments on ablating the factors directly onto the large-scale-trained models and test their correlation as done in Fig. 2 should be included.
- Legend colors do not match to those in Figure 9 (b).
- Please more provide details of Section 4.2., such as, what models authors compare, and what models the different colors in Fig. 10 indicate.

**Limitations:**

See weakness.

---

> ### Author Rebuttal · Authors · 2024-08-06
>
> Dear reviewer XjUF,
>
> Thank you so much for your thorough review and kind suggestions! We answer your questions as follows:
>
> > W1&Q2: Given large-scale data, how ineffective are other factors studied in the paper; Fig. 2 on large-scale datasets should be included
>
> Nice suggestion! We agree that scaling up data is one of the most effective factors, but it is only helpful for CLIP, and not for SL. Also, scaling up data could introduce heavy-tailed data distribution, and models can be thus more biased. To present a more complete picture, we also conducted the experiments in Fig. 2 on LAIONet and YFCC. The results are as follows:
>
> | Corr. (freq. vs. acc.)  | IN-Caps | LAIONet (match freq.) | LAIONet (full) | YFCC-15M | LAION-400M |
> | ----------------------- | :-----: | :-------------------: | :------------: | :------: | :--------: |
> | SL                      |  0.52   |         0.44          |      0.63      |   0.64   |     -      |
> | SL w/ sub. voc.         |  0.39   |         0.27          |      0.43      |   0.44   |     -      |
> | CLIP w/ class templates |  0.50   |         0.34          |      0.43      |   0.36   |     -      |
> | CLIP w/ captions        |  0.28   |         0.26          |      0.40      |   0.34   |    0.12    |
>
> LAIONet variants are as in our paper: the match-freq version matches the scale (449K) and class distribution of IN-Caps, and the full version is of high imbalance and larger scale (3.26M). Unlike other web data like YFCC, LAIONet is filtered to consist of 1K ImageNet classes.
>
> 1. Comparing between rows, the model-side factor (sub. voc.) is consistently effective, and the benefit of descriptive captions gradually narrows as data scales up sufficiently.
> 2. Comparing IN-Caps and LAIONet (match freq.), the distribution shift forms a regularization and all models are less biased.
> 3. Comparing LAIONet (match freq.) and LAIONet (full), despite scaling up from 0.45M to 3.3M, more severe data imbalance (see Fig. 2 right) makes all models more biased.
> 4. Comparing LAIONet (full), YFCC-15M, and LAION-400M, CLIP consistently benefits as data scales up, while SL cannot. This indicates to benefit from data factors, model factor is also important.
>
> > W2: Details of Fig. 5 are needed; how are results calculated and what is the take-away message?
>
> Thanks for pointing this out and we have polished Fig. 5 to make it more reader-friendly, please see the attached PDF (Fig. R1). The takeaway message is: distribution shift and higher image diversity in web data help improve model robustness, while extreme data imbalance in them harms robustness.
>
> > W3: Fig. 6 shows a counter-trend to our claim
>
> Thanks for reminding us! We checked with the plotting code and identified a bug in indexing. We have updated the figure in the attached PDF.
>
> > W4: Additional results on SL using rand init head + sub voc. is needed for fair comparison
>
> Please note that the existing comparison with frozen head is already a fair comparison and shows the effectiveness of vocabulary subsampling. Under the 0-shot tail (open-world) setting, a model without a pre-trained head has no knowledge about the mapping to novel classes, and thus cannot perform better than random guessing. We tried the setting suggested by the reviewer, and the best results in tail classes were smaller than 1% accuracy. Despite this, we expect the strategy to work under normal (non-extreme) data imbalance settings, eg, as shown in Fig. 3b, comparing darkest and lighter crosses.
>
> > W5&Q4: Setup of Sec. 4.2 need further clarifications
>
> Thanks for pointing this out. We provided more detailed settings of Sec. 4.2 in Sec. D, especially D.2. We will add a reference to the appendix to make it clearer. We compare DINO variants trained on LAIONet to the original DINO trained on ImageNet, in terms of transfer learning performance (also see Tab. 2 on page 22). Orange refers to original/vanilla DINO, and blue refers to DINO+vocabulary subsampling. A lighter color refers to a smaller prototype number.
>
> > Q1: How do we obtain ImageNet class frequency in web-scale datasets
>
> We provided details about the estimation process in Sec. B.2, and the models considered in Sec. B.3. We implemented a simple set-based substring matching method for efficient estimation on large-scale datasets. Despite being simple, we find it strongly correlates to results estimated using more advanced techniques (discussed in Sec. A.4).
>
> > Q3: Legend colors do not match to those in Fig. 9 (b)
>
> We have revised this figure following your suggestion (see attached PDF), thanks!

---

> > ### Comment · Reviewer_XjUF · 2024-08-11
> >
> > I appreciate the authors’ rebuttal. Please find my responses below:
> >
> > 1. My major concern in W1 and Q1 was that most of the analysis, e.g., Figs 4-b, 5-b and 8, using “Frozen CLIP head” taken from pretrained model with large-scale data is the dominant factor that provides the low correlations, while other factors show minor impacts. Specifically, in Fig.4-b, only those with Frozen CLIP head shows meaningfully low correlations, while others fail to do so. In addition, in Fig 8, results with "Frozen CLIP head" show improved tail accuracy, while others fail. These results give an impression that leveraging the "Frozen CLIP head" is the major factor that suggests the robustness among various factors explored in the paper, which is not surprising. The authors' responses do not seem to fully address the issue.
> > 2. Is “no distribution shift” equal to “=freq”?
> > 4. Thanks for the clarification. However, the authors' responses further strengthen my impression that this paper’s main contribution is heavily relying on large-scale pretrained CLIP head for heavy-tailed classes in both 1 and 0-shot.
> > 5. The authors' responses do not include the results of correlation, which is the main focus of this paper.
> > 3, 6, 7. Thanks for the clarification.

---

> ### Author Response · Authors · 2024-08-11
>
> Dear reviewer XjUF,
>
> Thanks for your responses and we would like to make some further clarifications:
>
> > Is "Frozen CLIP head" taken from pretrained model with large-sacle data the dominant factor?
>
> Concerning whether conclusions of Fig. 4b are dominated by “frozen CLIP head”:
>
> 1. It is important to emphasize that directly comparing the correlation values of models trained from scratch with those of a frozen pre-trained CLIP head is not fair, nor do we intend to make such a comparison in the figure. This is because the CLIP head is trained on 400M data points, while the other methods are trained on at most 3M data points. Due to limited computational resources, we could only scale our data to the million level for these studies. However, in Fig. 4b, we observe a clear trend: as the data scale increases, the correlation consistently decreases, indicating that data scale is indeed a significant factor in robustness.
> 2. Furthermore, in Fig. 1, where we scale the data from 3M to 2.5B, it becomes even more apparent that increasing the data amount leads to lower correlation values. It's important to note that *all models in Fig. 1 are trained from scratch*, meaning that the pre-trained head is not a contributing factor to the results in Fig. 1. Interestingly, the low correlation observed when using a frozen CLIP pre-trained model actually supports our hypothesis that robustness improves significantly with larger data scales. This implies that with substantial amounts of data, the trend toward increased robustness becomes more pronounced.
> 3. The reason we incorporated frozen CLIP head into Fig. 4 is to illustrate that the feature space learned by CLIP from large-scale data can serve as a teacher to finetune models with a smaller scale of training data while achieving robustness *for free*. Still, even in this setting, it can be observed that robustness can be enhanced with more training data.
>
> Concerning whether conclusions of Fig. 5b are dominated by “frozen CLIP head”:
>
> 4. In Fig. 5b, we use "frozen CLIP head" to 1) estimate the behaviors of a large-scale trained CLIP when only small-scale controlled data is available, and 2) verify how effective the data factors are given knowledge from large-scale data (exactly your major concern). The results from the "frozen CLIP head" and the "from scratch" setting should not be compared directly. Instead, they should be interpreted separately, as both approaches can independently verify the effect of data imbalance, diversity, and distribution shift. The results of Fig 5b show that tendencies of "from scratch" and "frozen CLIP head" are coherent, indicating these data factors are consistently effective, regardless of whether knowledge from large-scale data is available. Fig. 3 under this setting is also discussed in Fig. 17.
>
> We apologize for the confusion and will revise the figure to clarify the presentation. Additionally, we will separate the study into individual figures to enhance clarity.
>
> > Is “no distribution shift” equal to “=freq”?
>
> Sorry for the confusion, the revised figure in our PDF response might be clearer than the original one. Results between LAIONet subsets have no distribution shift, and only difference in intra-class image diversity (if using different thresholds) or level of data imbalance (if using same threshold, different class distribution). "=freq" refers to the LAIONet variant that has the same class distribution with IN-Caps, it should be compared with IN-Caps and the only difference between them is distribution shift.
>
> > For heavy-tailed classes in both 1 and 0-shot, is relying on large-scale pretrained CLIP head our main contribution?
>
> No, using pre-trained "frozen CLIP head" is already a common practice in open-world recognition (more similar to the setting in Sec. 4.1) and long-tailed classification, and not our novel contribution. But note that, none of them considered introducing the training mechanisms of CLIP (vocabulary subsampling) to the downstream task. Our contribution is to show adding vocabulary subsampling is important to acquire CLIP knowledge when its head is reused.
>
> > The authors' responses do not include the results of correlation
>
> Please note that Sec. 4.1 considers a very different setting than Sec. 3, aiming to let models better acquire CLIP knowledge when pre-trained CLIP head is used, and not to study factors behind CLIP's robustness. In this setting, the expected property is higher tail/novel-class accuracy. Correlation values indeed can be computed, but note that many classes have the same 1/0 frequency, and the correlation is less indicative in this case.

---

> > ### Comment · Reviewer_XjUF · 2024-08-11
> >
> > I greatly appreciate the authors' detailed responses for further clarification.
> >
> > - The authors' responses clarified their intuition that several data factors show trends of improving robustness when independently considering models trained from "scratch"  from those with"Frozen CLIP Head". However, I still think it is true that "Frozen CLIP Head" w/o considering such data factors (e.g., sub voc, data scale) even provides substantial improvements to robustness compared to "scratch" with considering such factors. In other words, the contribution of such factors becomes highly marginal when comparing to that of "Frozen CLIP Head". This takes back to my original concern: _using “Frozen CLIP head” taken from pretrained model with large-scale data is the dominant factor that provides the low correlations_.
> > - I greatly appreciate the authors' attempts for better understandability of the paper. However, I think the current paper requires more than minor level of revision for concise and clear presentation of the results as well as their insights, including critical error in Fig. 6 of the original submission.
> >
> > Therefore, I lean to maintaining my original rating for now, but I may re-assess after discussions with other reviewers.

---

> > > ### Author Response · Authors · 2024-08-14
> > > **Message from the Authors**
> > >
> > > Dear Reviewer XjUF,
> > >
> > > We sincerely appreciate the considerable time and effort you have dedicated to reviewing our paper. Your thought-provoking questions and insightful feedback have been instrumental in improving the quality of our work.
> > >
> > > We believe we have provided sufficient evidence to address your concerns. Although you still have concerns about the CLIP head (**not our contribution but just an analysis tool**), we have done our best to address it by showing additional experiments and explaining the figures. But before elaborating on this issue, we would like to re-emphasize the main focus and contributions of the paper, as the discussions above might have fallen too much into detail and lost the big picture.
> > >
> > > **What is this paper mainly about.**
> > > We are one of the first works to dig into the concept distribution of web-crawled datasets and reveal the robustness of industry-trained CLIP to it. We conduct a series of systemic and controlled experiments to ablate different factors and identify the influential ones to CLIP. We further verify that these factors can be transferred to SL/SSL settings. Given that the community is switching from model-centric to data-centric, from lab-scale data to web-scale data, our work 1) identifies a key problem: web data is extremely imbalanced; 2) identifies a good property of CLIP: robustness to such pre-train data; 3) reveals mechanisms behind it: both model and data factors; and 4) provide transferrable insights to other domains (eg, both long-tail and open-world recognition works have been using CLIP head as is, and we highlight that introducing its pre-training pretext to downstream task is also necessary). We also provided all code and data to the AC, believing that releasing this project could greatly benefit future related research.
> > >
> > > **The scaling law between data and robustness to data imbalance is one privilege of CLIP.**
> > > While it is commonly believed that scaling up data is beneficial to model performance (eg, accuracy), this does not directly apply to models' robustness to data imbalance/bias. One example is the results of SL (as in Fig. 1b). More importantly, **it also does not apply to generative LLM/MLLMs that are also pre-trained on massive data** (detailed in our response to reviewer tULz Q1). Considering their vocabulary as a giant class set, the way they predict each token (autoregressively) is also the same way with SL (using CE loss) and could be a cause of bias. Instead, our work reveals that CLIP implicitly applies subsampling to its vocabulary and continues to benefit from larger-scale data. This means **CLIP's factors are not undermined by, but are the root causes of more-data-more-robustness to data imbalance**.
> > >
> > > **Scaling up data is orthogonal to other factors and they can be mutual-complementary.**
> > > We also would like to emphasize that the factors we discovered and not contradictory with the data scale, and this is also exactly the reason we included results using frozen CLIP heads (which means knowledge from large-scale data). 1) In the additional experiment we provided in the rebuttal, we extended Fig. 3 to the largest scale we could, showing that caption diversity and vocabulary subsampling are consistently helpful as data scales up. 2) The blue lines (w/ frozen CLIP head) are aimed at **showing model behavior when large-scale knowledge is readily available and only small-scale controlled data is at hand**, and the trends cohere with training-from-scratch results, parallelly defending our findings. 3) The gap between this setting is apparent, but note that these factors are **mutual-complementary** with data scale -- **imagine what happens if we synthesize a giant dataset with extreme data imbalance and massive duplicated low-quality data** -- representations learned could be worse than training on CIFAR10.
> > >
> > > We sincerely hope that our responses have satisfactorily addressed your concerns. We remain open and enthusiastic about addressing any additional questions you may have, as your continued input would greatly contribute to further improving our work. We would like to thank you again for your devotion to reviewing our manuscript and will continue to improve its quality by all means. We also treasure this opportunity to share the problems found, the design of the study, and the findings with the NeurIPS community and sincerely hope these could be inspiring for future research.
> > >
> > > Best regards,
> > >
> > > Paper 4735 Authors

---

> ### Author Response · Authors · 2024-08-12
>
> Dear reviewer XjUF,
>
> Thanks for your great efforts in reviewing our paper. We respect your opinions but thought it best to emphasize that:
> - The aim of our paper is to answer **what makes existing industry-trained CLIP models robust to the data imbalance in their pre-training datasets**. We considered a total of 41 CLIP models in OpenCLIP, and their trend is presented in Fig. 1b as a motivating point. All of them are *trained from scratch*. Thus we already presented a complete study with the "from scratch" results. The head is not a valid factor to answer this question, as again, **all of them are trained from scratch**.
> - The reason we additionally provided a parallel discussion on the head is simply to 1) show a practical trick for training CLIP on small-scale data, and 2) answer the question "When CLIP is trained with much more data, how valid are other factors?" by simulating large-scale trained-from-scratch CLIPs. Combining with the former, the takeaways of our paper include both tricks for training CLIP with restricted computation and more general guidance. For the latter, our study re-evaluates the factors studied and shows that they are still valid when the model has more knowledge.
> - The fact that correlations with the head are much lower is expected -- it knows 400M data, compared to the 3M (at most) of training from scratch. Still, it is higher than that of Fig. 1b (trained from scratch on 400M data). When comparing from scratch and results with the head, control of variables is important. Again, **the head is not a valid factor of CLIP, and we just intended to show a practical trick to the community.** It is a very minor part of our paper and we also do not claim this trick as our contribution.

---

### Official Review · Reviewer_tULz · 2024-07-27

**Soundness:** 4
**Presentation:** 4
**Contribution:** 4
**Rating:** 7
**Confidence:** 3

**Summary:**

CLIP models are known to exhibit good robustness and high quality representations compared to supervised models. In this paper, the authors study CLIP models through the lens of the distribution of concepts in the underlying training datasets. The authors find that CLIP appears to demonstrate better robustness to data imbalance compared to supervised models. The authors conduct a detailed study the analyze various factors influencing robustness, and show that applying a dynamic classification task inspired by CLIP training improves robustness in supervised models.

**Strengths:**

- The paper is well motivated and original. The unusual robustness of CLIP is well-known, but its mechanism are not well understood
- The problem is clearly stated and the experimental analysis is clear.
- The datasets used are well-motivated and likely provide high quality data (especially ImageNet-Captions) for their experiments
- The experimental setup is sound, i.e. models are trained with the same backbones, data augmentation, and standard training techniques.
- Figures are presented well
- The paper is overall well-written

**Weaknesses:**

- A major experimental claim is that CLIP models exhibit a lower correlation between ImageNet class accuracy and frequency. However, this claim makes the assumption that the frequency of these classes in the training data is accurately measured, which may not be true for LAIONet, which is too large to verify. If rare classes are undercounted, which could affect the correlation.
- (Line 140) The statement alf a batch size 1024 training run having something like 600 imagenet classes is probably roughly true, although large models trained in industry clusters likely have a much larger batch size, and closer to 1000 classes.
- When applying the proposed class sub-sampling training method (Figure 10, section 4.2), the results are only stated qualitatively as narrowing the gap. However, a quantitative analysis would be more appropriate. In general, this proposed training technique seems very interesting, and I would appreciate seeing more significant analysis of how this could be used.
- The authors state that their method “uncovers the mechanisms behind CLIP’s generalizability’ (Line 12). However, while I agree that their experiments find quality evidence for several key relationships that are involved in generalizability/robustness, it is too generous to say that they have fully uncovered these mechanisms, or that they have been characterized in detail. I believe the authors should soften these claims.

**Questions:**

This paper describes several of the key mechanisms that influence model robustness. Besides the proposed training technique for supervised models, what recommendations would the authors propose for improving the next generation of self-supervised/CLIP models?

**Limitations:**

The authors have a section that describes the limitations of the work (Appendix F)

---

> ### Author Rebuttal · Authors · 2024-08-06
>
> Dear reviewer tULz,
>
> Thank you for your time and efforts in helping us improve our paper! We hereby answer your questions and resolve your concerns as follows:
>
> > W1: Frequency of classes in LAIONet may not be accurately measured
>
> 1. Thanks for pointing this out! This is why we tried to study as many factors on IN-Caps as possible. However, it is still expected that we verify these factors on other datasets to better support our findings (as requested by Reviewer XjUF), and provide a complete discussion on factors not supported by IN-Caps (eg, data distribution and data scale), thus facing an inevitable trade-off. Accurate concept frequency estimation of web datasets is an emerging topic and we believe it is valuable for future research.
> 2. Regarding LAIONet, the authors are aware of the label noise and explored applying a filter. They empirically found this to be effective in overcoming such noise (see Fig. 11 in LAIONet’s arXiv preprint). When replicating LAIONet, we are conservative on the choice of the filtering threshold, trying to make the selected images have highly confident labels.
> 3. We also explored an even simpler strategy: set-based substring matching (detailed in Sec. B.2). We applied this strategy to estimate the frequency in large-scale datasets (eg, LAION-400M). In comparison to advanced estimation techniques using ChatGPT and Llama2 (Parashar et al.), our method shows very consistent results (correlation of 0.8-0.9, see Sec. A.4 for details).
>
> > W2: Industry-trained CLIP have much larger batch sizes, thus all imagenet classes might be covered
>
> Please note that the industry trains on web datasets that cover way more concepts. Our setting of batch size 1024 considers IN-Caps that cover 1K concepts. LAION-400M could cover tens of thousands of long-tail distributed concepts. When sampling a batch of 32K images from these 400M images, it is likely that only a portion of 1K ImageNet classes would be retrieved.
>
> > W3: Quantitative analysis of Fig. 10 would be interesting
>
> 1. Thanks for acknowledging this analysis! For numbers in this figure before subtraction, please check Tab. 2 on page 22. Regarding the appropriate choice of vocabulary size, this figure suggests there is a sweet spot smaller than the total prototype number (eg, around 16384 for 65536 prototypes). Also, the lighter orange results (using 16384 prototypes w/o subsampling) show worse results, verifying the effectiveness of our strategy.
> 2. We recently noticed a study [a] that shows the learnable prototypes of DINO (akin to the classifier of SL) may be biased to imbalanced data and many of them collapse. Our strategy might help alleviate such collapse and encourage a more balanced utilization of these prototypes. We agree this finding is interesting and valuable for future research in clustering-based SSL, including SwAV, DINO, iBOT, DINOv2, etc.
> 4. We also provide DINO results trained on IN-Caps in the attached PDF (Fig. R6) to better support the findings.
> 5. We also would like to emphasize that our experiments of vocabulary subsampling are intended to verify the underlying rebalancing/debiasing mechanism of CLIP can be transferred to, other domains (SL and SSL). We do not intend to propose a novel method, but to point out problems to solve (model-side debiasing is widely needed when training on web data). We call for, and it is very likely better methods for debiasing large-scale trained SSL/LLM/MLLM to be developed in the future.
>
> Reference:
>
> [a] Govindarajan et al., On Partial Prototype Collapse in Clustering-Based Self-Supervised Learning
>
> > W4: One claim should be softened
>
> Thanks for reminding us! This sentence was intended to emphasize we study CLIP’s robustness/generalization to a specific property – data imbalance. We realize that this claim can be vague and strong, and will soften the statement and reduce the use of “generalization”. Also, we are considering to simplify the title to “What makes CLIP more robust to pre-training data imbalance”.
>
> > Q1: Recommendations for next-gen SSL/CLIP models
>
> In contrast to most previous efforts that study CLIP’s robustness to OOD evaluation data, we focus on long-tailed pre-training data. From this point, we indeed have some general messages to share with the community:
>
> 1. Data matters. Agreeing with a common trend in ML research, improving the quality of training data is as important as designing better models. Scaling up the data also helps.
> 2. Data is not the silver bullet. Robustness does not always come for free as data scales, and model-side debiasing is needed. CLIP can learn robust features from long-tailed data, and the robustness consistently improves as data scales up. In contrast, SL is always clearly biased despite being trained on larger-scale datasets.
> 3. Robustness to pre-training data imbalance is also something lacking in modern LLM/MLLMs, despite being trained on massive-scale data.
>    1. In Sec. 10 of [b], the authors show that if 1/8 “useful data” is mixed with 7/8 “junk data”, LLM model capacity degrades by 20x, and teaching the model to identify them effectively improves performance. Despite there are extensive efforts in data curation for LLMs (eg, data rebalancing), the large data corpus still can be long-tailed/biased, and more attention is needed.
>    2. In Fig. 3 of [c], the authors show that LLaVA’s poor classification performance is strongly correlated to the class frequency in training data, while not for CLIP.
>    3. These two findings indicate that CLIP’s robustness to pre-training data imbalance might not apply to generative models.
>
> Reference:
>
> [b] Allen-Zhu et al., Physics of Language Models: Part 3.3, Knowledge Capacity Scaling Laws
>
> [c] Zhang et al., Why are Visually-Grounded Language Models Bad at Image Classification?

---

> > ### Author Response · Authors · 2024-08-13
> > **Looking forward to having further discussions**
> >
> > Dear Reviewer tULz,
> >
> > Thank you so much for your time and efforts in reviewing our paper. Hope our rebuttal has addressed your concerns, and we will revise our paper based on your feedback. We are happy to discuss with you further if you have other concerns. Thanks for your time!
> >
> > Best regards,
> >
> > Paper 4735 authors

---

> > ### Comment · Reviewer_tULz · 2024-08-14
> > **Thank you for your response.**
> >
> > Thank you for addressing in depth my feedback for the paper. I believe that the response satisfies my points.

---

> > > ### Author Response · Authors · 2024-08-14
> > > **Thank you for your positive feedback!**
> > >
> > > Dear Reviewer tULz,
> > >
> > > We appreciate your positive feedback and are truly delighted to see that our response has addressed your concerns and questions. If all weakness points are resolved, would you mind considering a raise of your recommendation?
> > >
> > > Thank you once again for dedicating your time and effort to reviewing our work and providing us with insightful suggestions!

---

### Author Rebuttal · Authors · 2024-08-06

We thank all reviewers for their time, valuable comments, and kind suggestions. We also appreciate that our work is recognized to have a "novel" observation [tULz, XjUF, iwVX], study a "critical" problem [XjUF, iwVX, WXxS], design "sound" [tULz] and "comprehensive" [WXxS] experiments, and present "very interesting" [tULz] findings that "have practical insights" [XjUF].

In the responses to each reviewer, we have tried our best to resolve misunderstandings, provide experimental support, and clarify details. These discussions and new results have inspired us to improve our work and revise unclear writings and figures. We will incorporate all these in the next version of our manuscript.

In the attached PDF, we provide some revised figures according to the suggestions. We also have sent an anonymized link to our code in a separate comment to the AC, and please contact the AC for access. The code contains all the details to replicate this project, ranging from concept frequency estimation, dataset curation and ablation, and training and evaluating SL/CLIP/DINO models and their variants considered. We also provided notebooks to replicate all figures present in the paper. We will make it public to facilitate future research.

Thank you again and looking forward to discussing with you in the next period!

---

### Decision · Program_Chairs · 2024-09-25

**Decision:**

Accept (poster)

**Comment:**

Though the paper has mixed reviews, the majority of reviewers recommend acceptance and appreciate the detailed analysis of the robustness of CLIP-style models relative to their training data distributions and with respect to robustness for imbalanced data in particular. The AC team agrees that this analysis adds value to the community, and thus recommend acceptance.